# A Quantitative Geometric Approach to Neural-Network Smoothness

**Zi Wang**
University of Wisconsin-Madison
zw@cs.wisc.edu

**Gautam Prakriya**
The Chinese University of Hong Kong
gautamprakriya@gmail.com

**Somesh Jha**
University of Wisconsin-Madison
jha@cs.wisc.edu

## Abstract

Fast and precise Lipschitz constant estimation of neural networks is an important task for deep learning. Researchers have recently found an intrinsic trade-off between the accuracy and smoothness of neural networks, so training a network with a loose Lipschitz constant estimation imposes a strong regularization, and can hurt the model accuracy significantly. In this work, we provide a unified theoretical framework, a quantitative geometric approach, to address the Lipschitz constant estimation. By adopting this framework, we can immediately obtain several theoretical results, including the computational hardness of Lipschitz constant estimation and its approximability. We implement the algorithms induced from this quantitative geometric approach, which are based on semidefinite programming (SDP).[1] Our empirical evaluation demonstrates that they are more scalable and precise than existing tools on Lipschitz constant estimation for $\ell_\infty$-perturbations. Furthermore, we also show their intricate relations with other recent SDP-based techniques, both theoretically and empirically. We believe that this unified quantitative geometric perspective can bring new insights and theoretical tools to the investigation of neural-network smoothness and robustness.

## 1  Introduction

The past decade has witnessed the unprecedented success of deep learning in many machine learning tasks (Krizhevsky et al., 2017; Mikolov et al., 2013). Despite the growing popularity of deep learning, researchers have also found that neural networks are very vulnerable to adversarial attacks (Szegedy et al., 2014; Goodfellow et al., 2015; Papernot et al., 2016). As a result, it is important to train neural networks that are robust against those attacks (Madry et al., 2018). In recent years, the deep learning community starts to focus on certifiably robust neural networks (Albarghouthi, 2021; Hein and Andriushchenko, 2017; Katz et al., 2017; Cohen et al., 2019; Raghunathan et al., 2018; Wang et al., 2022; Leino et al., 2021). One way to achieve certified robustness is to estimate the smoothness of neural networks, where the smoothness is measured by the Lipschitz constant of the neural network. Recent works have found that to achieve both high accuracy and low Lipschitzness, the network has to significantly increase the model capacity (Bubeck and Sellke, 2021). This implies that there is an intrinsic tension between the accuracy and smoothness of neural networks.

Commonly considered adversarial attacks are the $\ell_\infty$ and $\ell_2$-perturbations in the input space. Leino et al. (2021); Cohen et al. (2019) have successfully trained networks with low $\ell_2$-Lipschitz constant,

---

[1]Our code is available at https://github.com/z1w/GeoLIP.

and Huang et al. (2021) trained networks with low local Lipschitzness for $\ell_\infty$-perturbations. There are few well-established techniques to train neural networks with low global Lipschitzness for $\ell_\infty$-perturbations. The techniques for the $\ell_2$-perturbation do not easily transfer to the $\ell_\infty$-case (Leino et al., 2021). One critical step is to measure the Lipschitz constant more precisely and efficiently. Jordan and Dimakis (2020) showed that for ReLU networks, it is NP-hard to approximate the Lipschitz constant for $\ell_\infty$-perturbations within a constant factor , and proposed an exponential-time algorithm to compute the exact Lipschitz constant. However, researchers are interested in more scalable approaches to certify and train networks. In this work, we consider the *Formal Global Lipschitz* constant (FGL) (See Equation (3)), which is roughly the maximum of the gradient operator norm, assuming all activation patterns on hidden layers are independent and possible. FGL is an upper bound of the exact Lipschitz constant and has been used in Raghunathan et al. (2018); Fazlyab et al. (2019); Latorre et al. (2020).

We address the Lipschitz constant estimation from the quantitative geometric perspective. Quantitative geometry aims to understand geometric structures via quantitative parameters, and has connections with many mathematical disciplines, such as functional analysis and combinatorics (Naor, 2013). In computer science, quantitative geometry plays a central role in understanding the approximability of discrete optimization problems. We approach those hard discrete optimization problems by considering the efficiently solvable continuous counterparts, and analyze the precision loss due to relaxation, which is often the SDP relaxation (Goemans and Williamson, 1995; Nesterov, 1998; Alon and Naor, 2004). The natural SDP relaxations for the intractable problems usually induce the optimal known polynomial time algorithms (Bhattiprolu et al., 2022). By adopting the quantitative geometric approach, we can immediately understand the computational hardness and approximability of FGL estimations. Our algorithms on two-layer networks are the natural SDP relaxations from the quantitative geometric perspective.

Latorre et al. (2020) employed polynomial optimization methods on the FGL estimation for $\ell_\infty$-perturbations. Polynomial optimization is a very general framework, and many problems can be cast in this framework (Motzkin and Straus, 1965; Goemans and Williamson, 1995). Therefore, we argue that this is not a precise characterization of the FGL-estimation problem. On the other hand, there are also several SDP-based techniques for FGL estimations. Raghunathan et al. (2018) proposed an SDP algorithm to estimate the FGL for $\ell_\infty$-perturbations of two layer networks, and Fazlyab et al. (2019) devised an SDP algorithm to estimate the $\ell_2$-FGL. We will demonstrate the intricate relationships between our algorithms and these existing SDPs on two-layer networks.

Several empirical studies have found that techniques on one $\ell_p$-perturbations often do not transfer to another ones, even though the authors claim that *in theory* these techniques should transfer (Fazlyab et al., 2019; Leino et al., 2021). This in-theory claim usually comes from a *qualitative* perspective. In finite-dimensional space, one can always bound one $\ell_p$-norm from another one, so techniques for one $\ell_p$-perturbations can also provide another bound for a different $\ell_p$-perturbations. However, this bound is loose and in practice not useful (Latorre et al., 2020). Instead, we believe that when transferring techniques from one norm to another one, we should consider the quantitative geometric principle: we should separate the geometry-dependent component from the geometry-independent one in those techniques, and modify the geometry-dependent component accordingly for a different normed space. As we will demonstrate, our whole work is guided by this principle. We hope that our unified quantitative geometric framework can bring insights to the empirical hard-to-transfer observations, and new tools to address these issues.

**Contributions.** To summarize, we have made the following contributions:

1. We provide a unified theoretical framework to address FGL estimations, which immediately yields the computational hardness and SDP-based approximation algorithms on two-layer networks (Section 3).

2. We demonstrate the relations between our algorithms and other SDP-based tools, which in return inspires us to design the algorithms for multi-layer networks. This provides more insightful and compositional interpretations of existing works, and makes them easier-to-generalize (Sections 4 and 5).

3. We implement the algorithms and name the tool GeoLIP. We empirically validate our theoretical claims, and compare GeoLIP with existing methods to estimate FGL for $\ell_\infty$-perturbations. The result shows that GeoLIP provides a tighter bound (20%-60% improvements) than

existing tools on small networks, and much better results than the naive matrix-norm-product method for deep networks, which existing tools cannot handle (Section 6).

## 2 Preliminaries

**Notation.** Let $[n] = \{1, \ldots, n\}$. For two functions $f$ and $g$, $f \circ g(x) = f(g(x))$. A 0-1 cube is $\{0, 1\}^n$, and a norm-1 cube is $\{-1, 1\}^n$ for some integer $n > 0$. $\mathbb{R}_+ = [0, \infty)$. For any vector $v \in \mathbb{R}^n$, $\mathrm{diag}(v)$ is an $n \times n$ diagonal matrix, with diagonal values $v$. Let $e_n = (1, \ldots, 1) \in \mathbb{R}^n$ be an $n$-dimensional vector of all 1's; and $I_n = \mathrm{diag}(e_n)$, the identity matrix. Let $||v||_p$ denote the $\ell_p$ norm of $v$. We use $q$ to denote the Hölder conjugate of $p$ as a convention, i.e., $\frac{1}{p} + \frac{1}{q} = 1$. If $v$ is an operator in the $\ell_p$-space, the operator norm of $v$ is then $||v||_q$. Throughout the paper, we consider the $\ell_p$-norm of the input's perturbation, and therefore, the $\ell_q$-norm of the gradient, which acts as an operator on the perturbation. A square matrix $X \succeq 0$ means that $X$ is positive semidefinite (PSD). Let $\mathrm{tr}(X)$ be the trace of a square matrix $X$. If $a, b \in \mathbb{R}^n$, let $\langle a, b \rangle$ be the inner product of $a$ and $b$.

**Lipschitz function.** Given two metric spaces $(X, d_X)$ and $(Y, d_Y)$, a function $f : X \to Y$ is *Lipschitz* continuous if there exists $K > 0$ such that for all $x_1, x_2 \in X$,

$$d_Y(f(x_2), f(x_1)) \leq K d_X(x_2, x_1). \tag{1}$$

The smallest such $K$ satisfying Equation (1), denoted by $K_f$, is called the Lipschitz constant of $f$. For neural networks, $X$ is in general $\mathbb{R}^m$ equipped with the $\ell_p$-norm. We will only consider the case when $Y = \mathbb{R}$. In actual applications such as a classification task, a neural network has multiple outputs. The prediction is the class with the maximum score. One can then use the margin between each pair of class predictions and its Lipschitz constant to certify the robustness of a given prediction (Raghunathan et al., 2018; Leino et al., 2021). From Rademacher's theorem, if $f$ is Lipschitz continuous, then $f$ is is almost everywhere differentiable, and $K_f = \sup_x ||\nabla f(x)||_q$.

**Neural network as function.** A neural network $f : \mathbb{R}^m \to \mathbb{R}$ is characterized by:

$$f_1(x) = W^1 x + b_1; \ f_i(x) = W^i \sigma(f_{i-1}(x)) + b_i, i = 2, \ldots, d.$$

where $W^i \in \mathbb{R}^{n_{i+1} \times n_i}$ is the weight matrix between the layers, $n_1 = m$, $d$ is the depth of the neural network, $\sigma$ denotes an activation function, $b_i \in \mathbb{R}^{n_{i+1}}$ is the bias term, and $f = f_d \circ \cdots \circ f_1$. Because we only consider the $\mathbb{R}$ as the codomain of $f$, $W^d = u \in \mathbb{R}^{1 \times n_d}$ is a vector. From chain rule, the gradient of this function is

$$\nabla f(x) = (W^1)^T [\mathrm{diag}(\sigma'(f_1(x)))(W^2)^T \cdots \mathrm{diag}(\sigma'(f_{d-1}(x)))(W^d)^T]. \tag{2}$$

Common activation functions, including ReLU (Nair and Hinton, 2010), sigmoid functions, and ELU (Clevert et al., 2016) are almost everywhere differentiable. As a result, we are interested in the supremum operator norm of Equation (2).

However, checking all possible inputs $x$ is infeasible, and common activation functions have bounded derivative values, say $[a, b]$. We are then interested in the following value instead:

$$\max_{v^i \in [a,b]^{n_i}} ||(W^1)^T \cdot \mathrm{diag}(v^2) \cdot \cdots \cdot \mathrm{diag}(v^d)(W^d)^T||_q, \tag{3}$$

where $n_i$ is the dimension of each $\mathrm{diag}(v^i)$. We call this value the *formal* global Lipschitz constant (FGL) because we treat all activation functions independent but in reality not all activation patterns are feasible. Therefore, this is an upper bound of the *true* global Lipschitz constant of the neural network. However, it is the value studied in most global Lipschitzness literature (Scaman and Virmaux, 2018; Fazlyab et al., 2019; Latorre et al., 2020), and also turns out useful in certifying the robustness of neural networks (Raghunathan et al., 2018; Leino et al., 2021; Pauli et al., 2022). We use $\ell_p$-FGL to denote the FGL for $\ell_p$-perturbations.

In this paper, we focus on the $\ell_p$-perturbation on the input, where $p = \infty$ or $p = 2$. Because $q$ is the Hölder conjugate of $p$, we are interested in the value of Equation (3), when $q = 1$ (for $p = \infty$), and $q = 2$ (for $p = 2$). Notice that in the ReLU-network case, $[a, b]$ is $[0, 1]$. We will use ReLU-networks as the illustration for the rest of the paper because of the popularity of ReLU in practice and the easy presentation of the 0-1-cube. However, the algorithms presented in this work can be adapted with minor adjustments to other common activation functions.

*Remark* 2.1. FGL considers all possible activation patterns on the hidden layers, while some of the activation patterns might be unachievable in reality. Therefore, FGL is an upper bound of the true Lipschitz constant. Notice that the activation pattern induced from an input is also decided by the bias term. Therefore, to find the true Lipschitz constant, one has to incorporate the information from the bias term.

# 3 Two-layer neural networks

In this section, we consider the two-layer neural network case. We reduce the FGL estimation to the matrix mixed-norm problem. This immediately yields the computational complexity and approximation algorithms for FGL estimations. In the appendix, we show that we can consider $\{0,1\}$ instead of $[0,1]$ in Equation (3) for two-layer networks.

**Problem description.** For a two-layer network where $W^1 = W \in \mathbb{R}^{n \times m}$ and $W^d = u \in \mathbb{R}^{1 \times n}$, its FGL (as in Equation (3)) is $\max_{y \in \{0,1\}^n} ||W^T \operatorname{diag}(y) u^T||_q$, where we use $y$ to denote $v^1$ in this case. If we expand the matrix multiplication, it is easy to check that this equals to $\max_{y \in \{0,1\}^n} ||W^T \operatorname{diag}(u) y||_q$. Let $A = W^T \operatorname{diag}(u)$, then the $\ell_p$-FGL is

$$\max_{y \in \{0,1\}^n} ||Ay||_q. \tag{4}$$

## 3.1 $\ell_\infty$-FGL estimation

We consider a natural SDP relaxation to Equation (4) when $q = 1$, and analyze the result using the celebrated *Grothendieck Inequality*, which is a fundamental tool in functional analysis.

**Mixed-norm problem.** The $\infty \to 1$ mixed-norm of a matrix is defined as

$$||A||_{\infty \to 1} = \max_{||x||_\infty = 1} ||Ax||_1.$$

The mixed-norm problem appears similar to Equation (4) when $q = 1$, except for that instead of a norm-1-cube, the cube in Equation (4) is a 0-1-cube. Alon and Naor (2004) showed that it is NP-hard, specifically MAXSNP-hard, to compute the $\infty \to 1$ mixed-norm of a matrix $A$, via a reduction to the graph Max-Cut problem. Moreover, Alon and Naor (2004) constructed a natural SDP relaxation for the mixed-norm problem:

$$\max \operatorname{tr}(BX)$$
$$s.t.\ X \succeq 0, X_{ii} = 1, i \in [n + m], \tag{5}$$

where $A$ is a submatrix of $B$. We provide the detailed derivation of this relaxation in the appendix. In fact, this relaxation admits a constant approximation factor. Grothendieck (1956) developed the local theory of Banach spaces, and showed that there exists an absolute value $K_G$ such that

**Theorem 3.1.** *For any $m, n \geq 1$, $A \in \mathbb{R}^{n \times m}$, and any Hilbert space $H$, the following holds:*

$$\max_{u_i, v_j \in B(H)} \sum_{i,j} A_{ij} \langle u_i, v_j \rangle_H \leq K_G ||A||_{\infty \to 1},$$

where $B(H)$ denotes the unit ball of the Hilbert space.

The precise value of $K_G$ is still an outstanding open problem, and it is known that $K_G < 1.783$ (Krivine, 1979; Braverman et al., 2011). The approximation factor of the SDP relaxation in Equation (5) is $K_G$. Similar to the mixed-norm problem, we show that the $\ell_\infty$-FGL estimation is MAXSNP-hard and provide an SDP relaxation, which also admits the $K_G$-approximation ratio. We provide a detailed explanation on why $K_G$ is the approximation ratio and how we can view the SDP relaxation as a geometric transformation in the appendix.

**Theorem 3.2.** $\ell_\infty$-*FGL estimation is* MAXSNP-*hard.*

**From 0-1 cube to norm-1 cube.** If we can transform the 0-1 cube in Equation (4) to a norm-1 cube, and formulate an equivalent optimization problem, then one can apply the SDP program in Equation (5) to compute an upper bound of the FGL. Indeed, we provide a cube rescaling technique,

and it allows us to construct the SDP for the $\ell_\infty$-FGL estimation. We provide the full detail of this technique in the appendix, and the result SDP for the $\ell_\infty$-FGL estimation is

$$\max \; \frac{1}{2}\mathrm{tr}(BX)$$
$$s.t. \; X \succeq 0, X_{ii} = 1, i \in [n+m+1], \tag{6}$$

where $B$ is a $(n+1+m) \times (n+1+m)$ matrix, and $B = \begin{pmatrix} 0 & 0 & 0 \\ A & Ae_n & 0 \end{pmatrix}$. As a result, we have:

**Theorem 3.3.** *There exists a polynomial-time approximation algorithm to estimate the $\ell_\infty$-FGL of two-layer neural networks, moreover, the approximation ratio is $K_G$.*

### 3.2 $\ell_2$-FGL estimation

Scaman and Virmaux (2018) showed that the $\ell_2$-FGL estimation is NP-hard. If $q = 2$ in Equation (4), the objective becomes similar to the $\infty \to 2$ mixed-norm problem. This is a quadratic optimization problem with a PSD weight matrix over a cube, and can be viewed as a generalization of the graph Max-Cut problem. In the quadratic-optimization formulation of Max-Cut, the weight matrix is the Laplacian of the graph, a special PSD matrix (Goemans and Williamson, 1995). Nesterov (1998) generalized Goemans-Williamson's technique and analyzed the case when the weight matrix is PSD, showing that the natural SDP relaxation in this case has a $\frac{\pi}{2}$-approximation ratio. This provides a $\sqrt{\frac{\pi}{2}}$-approximation algorithm for the $\ell_2$-FGL estimation. The approximation ratio comes from a similar inequality to the one in Theorem 3.1, known as the *Little Grothendieck Inequality*. The SDP for $\ell_2$-FGL estimation is:

$$\max \; \frac{1}{2}\sqrt{\mathrm{tr}(\begin{pmatrix} A^T A & A^T A e_n \\ e_n^T A^T A & e_n^T A^T A e_n \end{pmatrix} X)}$$
$$s.t. \; X \succeq 0, X_{ii} = 1, i \in [n+1]. \tag{7}$$

The full derivation is provided in the appendix, and we have the following theorem:

**Theorem 3.4.** *There exists a polynomial-time approximation algorithm to estimate the $\ell_2$-FGL of two-layer neural networks with an approximation factor $\sqrt{\frac{\pi}{2}}$.*

*Remark* 3.5. As we have discussed, for two-layer networks, the $\ell_p$-FGL estimation is essentially the $\infty \to q$ mixed-norm problem. Indeed the mixed-norm problem is an outstanding topic in theoretical computer science. As discussed in Bhattiprolu et al. (2018), the $\infty \to q$ mixed norm problem has constant approximation algorithms if $q \leq 2$, and is hard to approximate within almost polynomial factors when $q > 2$. Because when $q > 2$, its Hölder conjugate $p < 2$. This implies that for two-layer networks, the FGL estimation can be much harder for $\ell_p$-perturbations when $p < 2$.

Briët et al. (2017) showed that it is NP-hard to approximate the $\infty \to 2$ mixed-norm problem better than $\sqrt{\frac{\pi}{2}}$. Raghavendra and Steurer (2009) proved that assuming the unique games conjecture (Khot, 2002), it is NP-hard to approximate the $\infty \to 1$ mixed-norm problem better than $K_G$. These optimal approximation ratios match our SDP relaxations for FGL estimations accordingly.

## 4 Relations to existing SDP works

Before introducing our approach for multi-layer networks, we first examine some existing SDP works on FGL estimations, and discuss their relationships with our natural SDP relaxations in Section 3.

$\ell_\infty$**-FGL estimation.** Raghunathan et al. (2018) formulated an SDP that only works for two-layer networks. Theirs is essentially the same as ours in Equation (6) (See the detailed comparison in the appendix). However, we provide a rigorous derivation and simpler formulation, and also a sound theoretical analysis of the bound, which illustrate more insights to this problem. Raghunathan et al. (2018) treated the SDP relaxation as a heuristic to a hard quadratic programming problem. We prove that this relaxation is not only a heuristic, but in fact induces an approximation algorithm with a tight bound.

$\ell_2$**-FGL estimation.** Fazlyab et al. (2019) proposed LipSDP, another SDP-based algorithm for the $\ell_2$-FGL estimation problem. Fazlyab et al. (2019) provided several variants of LipSDP to balance the

precision and scalability. Pauli et al. (2022) demonstrated that the most precise version of LipSDP, *LipSDP-Network*, fails to produce an upper bound for $\ell_2$-FGL. In this paper, all the references of LipSDP are to *LipSDP-Neuron*, the less precise version. Surprisingly, even though the approach in LipSDP appears quite different from Equation (7), we show that LipSDP is dual of Equation (7) to estimate the $\ell_2$-FGL on two-layer networks. LipSDP for two-layer networks is:

$$\min_{\zeta,\lambda}\left\{\sqrt{\zeta}: \begin{pmatrix} -2abW^TWT - \zeta I_m & (a+b)W^TT \\ (a+b)TW & -2T + u^Tu \end{pmatrix} \preceq 0, \lambda_i \geq 0\right\},$$

where $T = \text{diag}(\lambda)$ for $\lambda \in \mathbb{R}^n_+$; $a$ and $b$ are the lower and upper bounds of the activation's derivative.

We will construct a new quadratic program, which we show is equivalent to Equation (4) when $q = 2$, and LipSDP is its dual SDP relaxation.

Let the input of the $i$-th activation node on $\text{diag}(y)$ be $y_i$, and $w_i$ be the row vector of $W$. Hence, $y_i = w_i x$. Let $\Delta x \in \mathbb{R}^m$ be a perturbation on $x$, so $\Delta y_i = w_i \Delta x$. Let $\Delta\sigma(y) \in \mathbb{R}^n$ denote the induced perturbation on $\text{diag}(y)$. The ***constraint*** from the activation function is $\frac{\Delta\sigma(y)_i}{\Delta y_i} \in [a, b]$, in other words, $\frac{\Delta\sigma(y)_i}{w_i \Delta x} \in [a, b]$. One can write the range constraint as

$$(\Delta\sigma(y)_i - a \cdot w_i \Delta x)(\Delta\sigma(y)_i - b \cdot w_i \Delta x) \leq 0.$$

This can be written in the quadratic form:

$$\begin{pmatrix} w_i \Delta x \\ \Delta\sigma(y)_i \end{pmatrix}^T \begin{pmatrix} -2ab & a+b \\ a+b & -2 \end{pmatrix} \begin{pmatrix} w_i \Delta x \\ \Delta\sigma(y)_i \end{pmatrix} \geq 0, \quad \forall i \in [n]. \tag{8}$$

Since for the two layer network, $f(x) = u\sigma(y)$, then $\Delta f(x) = u\Delta\sigma(y)$. The ***objective*** for $\ell_2$-FGL estimation is $\max_{\Delta x, \Delta\sigma(y)} \sqrt{\frac{(u\Delta\sigma(y))^2}{(\Delta x)^2}}$. The equivalence between this program and Equation (4) when $q = 2$ is presented in the appendix.

*Remark* 4.1. Another interpretation for the quadratic program is that we want to quantify how the output changes given a data-independent input change, i.e., $\Delta x$. In other words, we want to analyze the effect of $\Delta x$ propagating from the input to the output, with symbolic values rather than actual inputs. The idea is similar to symbolic execution from program analysis (Baldoni et al., 2018).

**Duality to LipSDP.** Now we will show that LipSDP is the dual SDP to the program formulated above. The dual SDP derivation is of similar form in Ben-Tal and Nemirovski (2001, Ch.4.3.1). Let us introduce a variable $\zeta$ such that $\zeta - \frac{(u\Delta\sigma(y))^2}{(\Delta x)^2} \geq 0$. In other words,

$$\zeta(\Delta x)^2 - (u\Delta\sigma(y))^2 \geq 0. \tag{9}$$

For each constraint in Equation (8), let us introduce a dual variable $\lambda_i \geq 0$. Multiply each constraint with $\lambda_i$, then $\begin{pmatrix} \Delta x \\ \Delta\sigma(y)_i \end{pmatrix}^T \begin{pmatrix} -2ab\lambda_i w_i^T w_i & (a+b)\lambda_i w_i^T \\ (a+b)\lambda_i w_i & -2\lambda_i \end{pmatrix} \begin{pmatrix} \Delta x \\ \Delta\sigma(y)_i \end{pmatrix} \geq 0, \quad \forall i \in [n].$

Sum all of them, then we have $\begin{pmatrix} \Delta x \\ \Delta\sigma(y) \end{pmatrix}^T \begin{pmatrix} -2abW^TWT & (a+b)W^TT \\ (a+b)TW & -2T \end{pmatrix} \begin{pmatrix} \Delta x \\ \Delta\sigma(y) \end{pmatrix} \geq 0$, where $T = \text{diag}(\lambda)$ is the $n \times n$ diagonal matrix of dual variables $\lambda_1, \ldots, \lambda_n$.

Equation (9) can be rewritten as: $\begin{pmatrix} \Delta x \\ \Delta\sigma(y) \end{pmatrix}^T \begin{pmatrix} \zeta I_m & 0 \\ 0 & -u^Tu \end{pmatrix} \begin{pmatrix} \Delta x \\ \Delta\sigma(y) \end{pmatrix} \geq 0.$ As a result, the dual program for the new optimization program is

$$\min_{\zeta,\lambda}\left\{\sqrt{\zeta}: \begin{pmatrix} -2abW^TWT - \zeta I_m & (a+b)W^TT \\ (a+b)TW & -2T + u^Tu \end{pmatrix} \preceq 0, \lambda_i \geq 0\right\}.$$

*Remark* 4.2. In Remark 3.5, we mention that $\sqrt{\frac{\pi}{2}}$ is the optimal approximation ratio for the $\infty \to 2$ mixed-norm problem, which matches the approximation ratio in Theorem 3.4. Hence, improving the natural SDP relaxation in Equation (7) can be very hard. The duality provides another evidence of LipSDP-Neuron's correctness, and hints that LipSDP-Network, the improved variant, may be wrong.

# 5  $\ell_\infty$-FGL estimation for multi-layer networks

For a multi-layer neural network, the formal gradient becomes a high-degree polynomial, and its $\ell_q$-norm estimation becomes a high-degree polynomial optimization problem over a cube, which is in general a hard problem (Lasserre, 2015). We provide a discussion of the polynomial optimization approach of FGL estimation in the appendix. Here we provide an SDP dual program of the $\ell_\infty$-FGL estimation inspired by the dual SDP approach in Section 4. The difference is that now we consider $\ell_\infty$-perturbations to the input space instead of $\ell_2$. Hence, the objective becomes

$$\max_{\Delta x, \Delta \sigma(y)} \frac{|u\Delta\sigma(y)|}{||\Delta x||_\infty}.$$

If we add an extra constraint $||\Delta x||_\infty = 1$, the above objective becomes

$$\max_{\Delta x, \Delta \sigma(y)} \frac{1}{2}(u\Delta\sigma(y) + u\Delta\sigma(y)). \tag{10}$$

The constraints are

$$\begin{pmatrix} w_i \Delta x \\ \Delta \sigma(y)_i \end{pmatrix}^T \begin{pmatrix} 2ab & -(a+b) \\ -(a+b) & 2 \end{pmatrix} \begin{pmatrix} w_i \Delta x \\ \Delta \sigma(y)_i \end{pmatrix} \leq 0, \ \Delta(x)_j^2 \leq 1, \ \forall i \in [n], j \in [m].$$

We can write $\Delta(x)_j^2 \leq 1$ as $\begin{pmatrix} 1 \\ \Delta x_j \end{pmatrix}^T \begin{pmatrix} 1 & 0 \\ 0 & -1 \end{pmatrix} \begin{pmatrix} 1 \\ \Delta x_j \end{pmatrix} \geq 0.$

Now let us introduce $n + m$ non-negative dual variables $(\tau, \lambda)$, where $\tau \in \mathbb{R}_+^m$ and $\lambda \in \mathbb{R}_+^n$. If we multiply each dual variable with the constraint and add all the constraints together, we will have

$$\begin{pmatrix} 1 \\ \Delta x \\ \Delta \sigma(y) \end{pmatrix}^T \begin{pmatrix} \sum_{j=1}^m \tau_j & 0 & 0 \\ 0 & -2abW^T W T_2 - T_1 & (a+b)W^T T_2 \\ 0 & (a+b)T_2 W & -2T_2 \end{pmatrix} \begin{pmatrix} 1 \\ \Delta x \\ \Delta \sigma(y) \end{pmatrix} \geq 0,$$

where $T_1 = \text{diag}(\tau)$ and $T_2 = \text{diag}(\lambda)$. As a result, we can incorporate the objective Equation (10) and obtain the dual SDP for the $\ell_\infty$-FGL estimation:

$$\min_{\zeta, \lambda, \tau} \left\{ \frac{\zeta}{2} : \begin{pmatrix} \sum_{j=1}^m \tau_j - \zeta & 0 & u \\ 0 & -2abW^T W T_2 - T_1 & (a+b)W^T T_2 \\ u^T & (a+b)T_2 W & -2T_2 \end{pmatrix} \preceq 0, \lambda_i, \tau_j \geq 0 \right\}. \tag{11}$$

*Remark* 5.1. The SDP programs in Section 3 are strictly feasible because the identity matrix is a positive definite solution. Hence, Slater's condition is satisfied and strong duality holds.

**Multi-layer extension.**  We can simply extend the dual program to multiple-layer networks. We first vectorize all the units in the input layer and hidden layers, and then constrain them using layer-wise inequalities to formulate an optimization problem. Let us consider a general $d$-layer multi-layer network, where $W^i \in \mathbb{R}^{n_{i+1} \times n_i}$ for $i \in [d-1]$, and $W^d = u \in \mathbb{R}^{1 \times n_d}$. Let $\Delta x$ denote the perturbation on the input layer, $\Delta z^i$ be the perturbation on the $i$-th hidden layer, and $w_j^i$ be the $j$-th row vector of $W^i$. The only difference between two layer networks and multi-layer networks is that we have the additional constraints:

$$\begin{pmatrix} \Delta z^i \\ \Delta z_j^{i+1} \end{pmatrix}^T \begin{pmatrix} -2ab(w_j^{i+1})^T w_j^{i+1} & (a+b)(w_j^{i+1})^T \\ (a+b)w_j^{i+1} & -2 \end{pmatrix} \begin{pmatrix} \Delta z^i \\ \Delta z_j^{i+1} \end{pmatrix} \geq 0.$$

Let $\Lambda_i \in \mathbb{R}_+^{n_i}$ and $T_i = \text{diag}(\Lambda_i)$ for $i \in [d]$. Following the similar SDP dual approach, we can add all the constraints together and formulate the following SDP program:

$$\min_{\zeta, \Lambda_i} \left\{ \frac{\zeta}{2} : (L + N) \preceq 0, i \in [d] \right\}, \tag{12}$$

where

$$L = \begin{pmatrix} 0 & 0 & 0 & \ldots & 0 & 0 \\ 0 & -2ab(W^1)^T W^1 T_2 & (a+b)(W^1)^T T_2 & \ldots & 0 & 0 \\ 0 & (a+b)T_2 W^1 & -2T_2 - 2ab(W^2)^T W^2 T_3 & \ldots & 0 & 0 \\ \vdots & \vdots & \vdots & \ddots & \vdots & \vdots \\ 0 & 0 & 0 & \ldots & (a+b)T_d W^{d-1} & -2T_d \end{pmatrix}, \tag{13}$$

Table 1: $\ell_\infty$-FGL estimation of various methods: DGeoLIP and NGeoLIP induce the same values on two layer networks. DGeoLIP always produces tighter estimations than LiPopt and MP do.

| Network | DGeoLIP | NGeoLIP | LiPopt | MP | Sample | BruF |
|---|---|---|---|---|---|---|
| 2-layer/16 units | 185.18 | 185.18 | 259.44 | 578.54 | 175.24 | 175.24 |
| 2-layer/256 units | 425.04 | 425.04 | 1011.65 | 2697.38 | 306.98 | N/A |
| 8-layer/64 units per layer | 8327.2 | —— | N/A | $8.237 * 10^7$ | 1130.6 | N/A |

Table 2: Running time (in seconds) of various tools on $\ell_\infty$-FGL estimation: DGeoLIP and NGeoLIP are faster than LiPopt. Notice that the running time is implementation and solver-dependent.

| Network | DGeoLIP | NGeoLIP | LiPopt | BruF |
|---|---|---|---|---|
| 2-layer/16 units | 28.1 | 22.3 | 1572 | 4.8 |
| 2-layer/256 units | 976.0 | 70.9 | 2690 | N/A |
| 8-layer/64 units | 329.5 | —— | N/A | N/A |

$$N = \begin{pmatrix} \sum_{k=1}^{n_1} \Lambda_{1k} - \zeta & 0 & \dots & u \\ 0 & -T_1 & \dots & 0 \\ \vdots & \vdots & \ddots & \vdots \\ u^T & 0 & \dots & 0 \end{pmatrix}.$$

*Remark* 5.2. If we expand the matrix inequality derived from the compact neural-network representation in Fazlyab et al. (2019, Theorem 2), we will have exactly the same matrix for network constraints as $L$ (Equation (13)) in the dual program formulation. In other words, we provide a compositional optimization interpretation to the compact neural-network representation in LipSDP. With this interpretation, one can extend the SDP to beyond feed-forward structures, such as skip connections (He et al., 2016). Notice that if we apply the similar reasoning to the multi-layer network $\ell_2$-FGL estimation, we will obtain LipSDP-Neuron.

## 6 Evaluation and discussion

The primary goal of our work is to provide a theoretical framework, and also algorithms for $\ell_\infty$-FGL estimations on practically used networks. The $\ell_2$-FGL can be computed using LipSDP. We have implemented the algorithms using MATLAB (MATLAB, 2021), the CVX toolbox (CVX Research, 2020) and MOSEK solver (ApS, 2019), and name the tool GeoLIP. To validate our theory and the applicability of our algorithms, we want to empirically answer the following research questions:

> **RQ1:** Is GeoLIP better than existing methods in terms of precision and scalability?
>
> **RQ2:** Are the dual SDP programs devised throughout the paper valid?

As we shall see, GeoLIP is indeed better than existing methods in terms of precision and scalability; and the dual SDP programs produce the same values as their natural-SDP-relaxation counterparts.

### 6.1 Experimental design

To answer **RQ1**, we will run GeoLIP and existing tools that measure the $\ell_\infty$-FGL on various feed-forward neural networks trained with the MNIST dataset (LeCun and Cortes, 2010). We will record the computed $\ell_\infty$-FGL to compare the precision, and the computation time to compare the scalability.

To answer **RQ2**, we will run the natural SDP relaxations for $\ell_p$-FGL estimations proposed in Section 3, LipSDP for $\ell_2$-FGL estimation, and the dual program Equation (11) for $\ell_\infty$-FGL on two-layer neural networks, and compare their computed FGLs.

**Measurements.** Our main baseline tool is ***LiPopt*** (Latorre et al., 2020), which is an $\ell_\infty$-FGL estimation tool.[2] Notice that LiPopt is based on the Python Gurobi solver (Gurobi Optimization, LLC, 2022), while we use the MATLAB CVX and MOSEK solver. LiPopt relies on a linear programming (LP) hierarchy for the polynomial optimization problem. We use LiPopt-k to denote the $k$-th degree of the LP hierarchy. ***BruF*** stands for an brute-force exhaustive enumeration of all possible activation patterns. This is the ground truth for FGL estimations. However, this is an exponential-time search, so we can only run it on networks with a few hidden units. ***Sample*** means that we randomly sample $200,000$ points in the input space and compute the gradient norm at those points. Notice that this is a lower bound of the true Lipschitz constant, and thus a lower bound of the FGL. ***MP*** stands for the weight-matrix-norm-product method. This is a naive upper bound of FGL. We use ***NGeoLIP*** to denote the natural SDP relaxations devised in Section 3, and ***DGeoLIP*** to denote the dual SDP Equation (12) for $\ell_\infty$-FGL estimation. Notice that NGeoLIP only applies to two-layer networks.

We use "—" in the result tables to denote that the experimental setting is not in the scope of the tool's application, and "N/A" to denote the computation takes too much time ($> 20$ hours).

**Network setting.** We run the experiments on fully-connected feed-forward neural networks, trained with the MNIST dataset for 10 epochs using the ADAM optimizer (Kingma and Ba, 2015). All the trained networks have accuracy greater than $92\%$ on the test data. For two-layer networks, we use 8, 16, 64, 128, 256 hidden nodes. For multiple-layer networks, we consider 3, 7, 8-layer networks, and each hidden layer has 64 ReLU units. Because MNIST has 10 classes, we report the estimated FGL with respect to label 8 as in Latorre et al. (2020), and the average running time per class: we record the total computation time for all 10 classes from each tool, and report the average time per class.

## 6.2 Discussion

We present selected results in Tables 1 and 2, and related major discussions here. The full results, more experimental setup and additional discussions can be found in the appendix.

**RQ1.** In the experiments of LiPopt, we only used LiPopt-2. In theory, if one can go higher in the LP hierarchy in LiPopt, the result becomes more precise. However, in the case of fully-connected neural networks, using degree-3 in LiPopt is already impractical. For example, on the simplest network that we used, i.e., the single-hidden-layer neural network with 8 hidden units, using LiPopt-3, one $\ell_\infty$-FGL computation needs at least 200 hours projected by LiPopt. As a result, for all the LiPopt-related experiments, we were only able to run LiPopt-2. As Latorre et al. (2020) pointed out, the degree has to be at least the depth of the network to compute a valid bound, so we have to use at least LiPopt-k for $k$-layer networks. LiPopt is unable to handle neural networks with more than two layers because this requires LiPopt with degrees beyond 2. Even if we only consider LiPopt-2 on two-layer networks, the running time is still much higher compared to GeoLIP. This demonstrates the great advantage of GeoLIP in terms of scalability compared with LiPopt. If we compare LiPopt-2 with GeoLIP on two-layer networks from Table 1, it is clear that GeoLIP produces more precise results. For networks with depth greater than 2, we can only compare GeoLIP with the matrix-norm-product method. As we can see from all experiments, GeoLIP's estimation on the FGL is always much lower than MP.

We have also shown that the two-layer network $\ell_\infty$-FGL estimation from GeoLIP has a theoretical guarantee with the approximation factor $K_G < 1.783$ (Theorem 3.3). If we compare the two-layer network results from GeoLIP and Sampling in Table 1, which is a lower bound of true Lipschitz constant, the ratio is within $1.783$. This validates our theoretical claim.

**RQ2.** In Section 4, we have demonstrated the duality between NGeoLIP and LipSDP for the $\ell_2$-FGL estimation on two-layer networks, even though the approaches appear drastically different. The experiments show that on two-layer networks, LipSDP and NGeoLIP for $\ell_2$-FGL estimations (in the appendix), and DGeoLIP and NGeoLIP for $\ell_\infty$-FGL estimations produce the same values. These results empirically validate the duality arguments, and also all the related SDP programs.

**SDP relaxation.** Applying SDP on intractable combinatorial optimization problem was pioneered by the seminal Goemans-Williamson algorithm for the Max-Cut problem (Goemans and Williamson, 1995). For two-layer networks, we have reduced the FGL estimation to the mixed-norm problem, and

---

[2]Another method was proposed by Chen et al. (2020), however, the code is not available and we are not able to compare it with GeoLIP.

provide approximation algorithms with ratios compatible with the known optimal constants in the corresponding mixed-norm problems. Improving them can be a very hard task. We also provide a compositional SDP interpretation of LipSDP-Neuron. Although Pauli et al. (2022) demonstrated the flaw in LipSDP-Network, our compositional SDP interpretation shows that LipSDP-Neuron is correct. In fact, from the compositional SDP interpretation, the program is only constrained by the underlying perturbation geometry and the layer-wise restriction from each hidden unit, so the constraints and objective exactly encode the FGL-estimation problem without additional assumptions. Because often the SDP relaxation for intractable problems gives the optimal known algorithms, we conjecture that GeoLIP and LipSDP are also hard to improve on FGL estimations.

Latorre et al. (2020) used polynomial optimization to address the $\ell_\infty$-FGL estimation. We argue that approaching FGL-estimations from the perspective of polynomial optimization loses the accurate characterization of this problem. For example, for two-layer networks, we have provided constant approximation algorithms to estimate FGLs in both $\ell_\infty$ and $\ell_2$ cases. However, for a general polynomial optimization problem on a cube, we cannot achieve constant approximation. For example, the maximum independent set of a graph can be encoded as a polynomial optimization problem over a cube (Motzkin and Straus, 1965), but the maximum independent set problem cannot be approximated within a constant factor in polynomial time unless $P = NP$ (Trevisan, 2004).

## 7   Related work

Chen et al. (2020) employed polynomial optimization to compute the true Lipschitz constant of ReLU-networks for $\ell_\infty$-perturbations, and used Lasserre's hierarchy of SDPs (Lasserre, 2001) to solve the polynomial optimization problem. However, their approach is highly tailored to ReLU networks, while ours, like LipSDP, can handle common activations, such as sigmoid and ELU.

Latorre et al. (2020) also proposed to use LiPopt to estimate the local Lipschitz constant. However, estimating this quantity is not the problem studied in our work, and there are tools specifically designed for local perturbations and the Lipschitz constant (Laurel et al., 2022; Zhang et al., 2019).

Lipschitz regularization of neural networks is an important task, and recent works (Aziznejad et al., 2020; Bungert et al., 2021; Gouk et al., 2021; Krishnan et al., 2020; Terjék, 2020) have investigated this problem. However, here we study a related but different problem, i.e., Lipschitzness measurement of neural networks. Our work can motivate new Lipschitz regularization techniques.

## 8   Conclusion

In this work, we have provided a quantitative geometric framework for FGL estimations, and also algorithms for the $\ell_\infty$-FGL estimation. One important lesson is that when transferring techniques from one perturbations to another ones, we should also transfer the underlying geometry. One future work is to train smooth neural networks using the SDPs proposed in this paper.

### Acknowledgments and Disclosure of Funding

The authors thank Vijay Bhattiprolu for introducing recent progress on the mixed-norm problems. The work is partially supported by Air Force Grant FA9550-18-1-0166, the National Science Foundation (NSF) Grants CCF-FMitF-1836978, IIS-2008559, SaTC-Frontiers-1804648, CCF-2046710 and CCF-1652140, and ARO grant number W911NF-17-1-0405. Zi Wang and Somesh Jha are partially supported by the DARPA-GARD problem under agreement number 885000.

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
