# A Elided background, derivations and proofs

## A.1 Additional analysis background

**Gradient as operator.** If a function $g : \mathbb{R}^m \to \mathbb{R}$ is a differentiable function at $a \in \mathbb{R}^m$, then the total derivative of $g$ at $a$ is

$$Dg(a) = [\frac{\partial g}{\partial x_1}(a), \dots, \frac{\partial g}{\partial x_m}(a)],$$

and the gradient of $g$ at $a$ is $\nabla g(a)$ is the transpose of $Dg(a)$. The linear approximation of $g$ at $a$ is $\langle Dg(a), dx \rangle$. Equivalently, we can view the change of a function with respect to an infinitesimal

perturbation as the inner product of $\nabla g(a)$ and $dx$. In this sense, the gradient acts as an operator on the perturbation.

**Differentiable activation.** Because we want to upper bound the true Lipschitz constant, we only need to show that the quantity considered in the paper indeed upper bounds the true Lipschitz constant considered in the paper. If the activation function is differentiable, then the neural network $f$ is also differentiable, so Equation (3) is trivially true, as proved and applied in Latorre et al. (2020, Theorem 1 and Equation 4).

**ReLU activation.** For ReLU networks, it is true if we have $[a, b] = [0, 1]$. One can consider the (Clarke) generalized Jacobian as in Jordan and Dimakis (2020). At each input point, the Clarke Jacobian is contained in $\{(W^1)^T \cdot \text{diag}(v^2) \cdot \cdots \cdot \text{diag}(v^d)(W^d)^T \mid v^i \in [a, b]^{n_i}\}$. Alternatively, we can also use the perturbation propagation argument in Section 4 to see this upper bound. Note that Raghunathan et al. (2018) used this interval representation for ReLU's derivative.

**Maximum over hypercube.** Now we want to show that the optimization problems over hypercubes considered in this work attain the maximum at the vertices. Without of loss of generality, let us assume the hypercube is $[-1, 1]^n$. Otherwise, we can transform the hypercube to $[-1, 1]^n$. Let $A \in \mathbb{R}^{m \times n}$, $x \in \mathbb{R}^n$, $y \in \mathbb{R}^m$, and $z \in \mathbb{R}^n$.

We will use the following facts

1. $||x||_1 = \max_{z \in \{-1,1\}^n} \langle x, z \rangle$;

2. $\ell_\infty$ is the dual of $\ell_1$;

3. Let $U \subseteq \mathbb{R}^n$. When $\max_{x,z \in U} \langle Ax, Az \rangle$ is well-defined, we have $\max_{x \in U} \langle Ax, Ax \rangle = \max_{x,z \in U} \langle Ax, Az \rangle$.

The first fact is from $||x||_1 = |x_1| + \ldots + |x_n| = \max_{z \in \{-1,1\}^n} \langle x, z \rangle$. The second fact is from Hölder's inequality for finite-dimensional vector space. For the third one, $\langle Ax, Az \rangle$ is maximized only when $Ax = Az$. Now we can show that the maximization problems considered in this paper attain the maximum at the hypercube vertices.

$$
\begin{aligned}
\max_{||x||_\infty = 1} ||Ax||_1 &= \max_{||x||_\infty = 1, y \in \{-1,1\}^m} \langle Ax, y \rangle && \text{(From fact (1))} \\
&= \max_{||x||_\infty = 1, y \in \{-1,1\}^m} \langle x, A^T y \rangle \\
&= \max_{y \in \{-1,1\}^m} ||A^T y||_1 && \text{(From fact (2))} \\
&= \max_{x \in \{-1,1\}^n, y \in \{-1,1\}^m} \langle x, A^T y \rangle \\
&= \max_{x \in \{-1,1\}^n, y \in \{-1,1\}^m} \langle Ax, y \rangle.
\end{aligned}
\tag{14}
$$

$$
\begin{aligned}
\max_{||x||_\infty = 1} ||Ax||_2^2 &= \max_{||x||_\infty = 1} \langle Ax, Ax \rangle \\
&= \max_{||x||_\infty = 1, ||z||_\infty = 1} \langle Ax, Az \rangle && \text{(From fact (3))} \\
&= \max_{||x||_\infty = 1, ||z||_\infty = 1} \langle A^T Ax, z \rangle.
\end{aligned}
$$

Using the similar idea in Equation (14), we have

$$
\max_{||x||_\infty = 1} ||Ax||_2^2 = \max_{x \in \{-1,1\}^n, z \in \{-1,1\}^n} \langle Ax, Az \rangle = \max_{x \in \{-1,1\}^n} \langle Ax, Ax \rangle.
$$

More generally, in the bilinear forms considered above, if $x = x_1 \otimes \cdots \otimes x_d$ is generated by the tensor product of variables over cubes, we can fix one variable and write $x$ as a matrix product, and then move the fixed variable to the hypercube vertices. We can repeat this process to move all variables to the vertices.

## A.2 Additional definitions

For any neural network $f$, let $OPT(f)$ be the optimal value of Equation (3). We say an algorithm $\mathcal{A}$ is an approximation algorithm for Equation (3) with approximation ratio $\alpha > 1$, if $OPT(f) \leq \mathcal{A}(f) \leq \alpha OPT(f)$.

## A.3 SDP for the $\infty \to 1$ mixed-norm problem

Recall that for $v \in \mathbb{R}^m$, $||v||_1 = \max_{||u||_\infty = 1} \langle u, v \rangle$. We can reformulate the mixed-norm problem as follows:

$$\max_{x \in \{-1,1\}^n} ||Ax||_1 = \max_{(x,y) \in \{-1,1\}^{n+m}} \langle Ax, y \rangle.$$

If we let $z = \begin{pmatrix} x^T & y^T \end{pmatrix}$, we can have

$$\max_{(x,y) \in \{-1,1\}^{n+m}} \langle Ax, y \rangle = \max_{z \in \{-1,1\}^{n+m}} z \cdot B \cdot z^T,$$

where $B$ is a $(m+n) \times (m+n)$ matrix. The last $m$ rows and first $n$ columns of $B$ is $A$, and the rest are 0: $B = \begin{pmatrix} 0 & 0 \\ A & 0 \end{pmatrix}$.

The natural SDP relaxation of the $\infty \to 1$ mixed-norm problem is:

$$\max \operatorname{tr}(BX)$$
$$s.t. \ X \succeq 0, X_{ii} = 1, i \in [n+m].$$

In other words, we treat $X$ as the SDP matrix relaxed from the rank-1 matrix $z^T \cdot z$.

## A.4 SDP relaxation and Grothendieck inequalities

In this work, we used the Grothendieck inequality as in Theorem 3.1:

$$\max_{u_i, v_j \in B(H)} \sum_{i,j} A_{ij} \langle u_i, v_j \rangle_H \leq K_G ||A||_{\infty \to 1}, \tag{15}$$

for any $A \in \mathbb{R}^{n \times m}$; and the little Grothendieck inequality:

$$\max_{u_i, v_j \in B(H)} \sum_{i,j} (A^T A)_{ij} \langle u_i, v_j \rangle_H \leq \frac{\pi}{2} ||A||_{\infty \to 2}^2. \tag{16}$$

Notice that $A^T A$ is a PSD matrix.

As discussed in Appendix A.3,

$$||A||_{\infty \to 1} = \max_{z \in \{-1,1\}^{n+m}} z \cdot B \cdot z^T.$$

The natural SDP relaxation is

$$\max \operatorname{tr}(BX)$$
$$s.t. \ X \succeq 0, X_{ii} = 1, i \in [n+m].$$

Because $X \succeq 0$, $X = MM^T$ for some $M \in \mathbb{R}^{(m+n) \times d}$, where $d \geq 1$. Let $M_i$ be the $i$-th row vector of $M$. $X_{ij} = \langle M_i, M_j \rangle$, and $X_{ii} = 1$ means $\langle M_i, M_i \rangle = 1$. As a result, $\operatorname{tr}(BX) = \sum_{i,j} A_{ij} X_{ij} = \sum_{i,j} A_{ij} \langle M_i, M_j \rangle_H$, where $H$ is the Hilbert space of $\mathbb{R}^d$ equipped with the canonical inner product. Thus, Equation (15) implies that $K_G$ is the approximation ratio in the SDP relaxation for the mixed-norm problem.

In contrast, in the mixed-norm problem, the variable to $B_{ij}$ is $z_i z_j$, the product of two scalars. If $d = 1$ in the SDP relaxation, $M$ is a column vector, and $X$ is a rank-1 matrix. In this case, the SDP coincides with the combinatorial problem, because the inner product degenerates to the multiplication of two scalars. Hence, the SDP relaxation can be viewed as a continuous relaxation of a discrete problem, and Equation (15) quantifies this geometric transformation. Another interpretation for the SDP relaxation is that SDP drops the rank-1 constraint in the quadratic formulation of the mixed-norm problem.

## A.5 Rescaling from 0-1 cube to norm-1 cube

Now let us show how we transform the 0-1 cube in Equation (4) to a norm-1 cube, and formulate an equivalent optimization problem. As a result, we can apply the SDP program in Equation (5) to compute an upper bound of the $\ell_\infty$-FGL.

Let $x_i = (t_i + 1)/2$, where $t_i \in \{-1, 1\}$. We have

$$\max_{x \in \{0,1\}^n} \|Ax\|_1$$

$$= \max_{x \in \{0,1\}^n, y \in \{-1,1\}^m} y^T A x \tag{17}$$

$$= \max_{t \in \{-1,1\}^n, y \in \{-1,1\}^m} \frac{1}{2} y^T A(t + e_n).$$

Let $OPT_1$ be the optimal value of

$$\max_{(t,y) \in \{-1,1\}^{n+m}} y^T A(t + e_n).$$

Introduce another variable $\tau \in \{-1, 1\}$, and let $OPT_2$ be the optimal value of

$$\max_{(t,y,\tau) \in \{-1,1\}^{n+m+1}} y^T A(t + \tau e_n). \tag{18}$$

**Lemma A.1.** $OPT_1 = OPT_2$.

*Proof.* Clearly $OPT_2 \geq OPT_1$.

Now if $(\hat{t}, \hat{y}, \tau = -1)$ is an optimal solution to Equation (18), then $(-\hat{t}, -\hat{y}, \tau = 1)$ is also an optimal solution, so $OPT_2 \leq OPT_1$. $\qquad\square$

Now let $z = (t, \tau)$, and we can verify that $y^T A(t + \tau e_n) = y^T B z$, where $B = (A \quad Ae_n)$.

As a result, the semidefinite program to the $\ell_\infty$-FGL constant is

$$\max \frac{1}{2} \text{tr}(BX)$$

$$s.t. \ X \succeq 0, X_{ii} = 1, i \in [n + m + 1],$$

where $B$ is a $(n + 1 + m) \times (n + 1 + m)$ matrix, and $B = \begin{pmatrix} 0 & 0 & 0 \\ A & Ae_n & 0 \end{pmatrix}$.

## A.6 Proof of Theorem 3.2

*Proof.* We will use the cube rescaling techniques introduced in Appendix A.5. Alon and Naor (2004) showed that matrix cut-norm is MAXSNP-hard. We will show that if one can solve the FGL estimation problem, then one can find the cut norm of a matrix.

Given a matrix $A$, the cut norm of a matrix $A \in \mathbb{R}^{m \times n}$ is defined as

$$CN(A) = \max_{x \in \{0,1\}^n, y \in \{0,1\}^m} \langle Ax, y \rangle.$$

We need to transform $y$ from 0-1 cube to norm-1 cube, so similarly let $y_i = (t_i + 1)/2$, where $t_i \in \{-1, 1\}$. The we will have

$$CN(A) = \max_{x \in \{0,1\}^n, y \in \{0,1\}^m} \langle Ax, y \rangle = \frac{1}{2} \max_{x \in \{0,1\}^n, t \in \{-1,1\}^m} \langle Ax, (t + e_m) \rangle.$$

Let $B = \begin{pmatrix} A \\ e_m^T A \end{pmatrix}$. From above we know that

$$\max_{x \in \{0,1\}^n, t \in \{-1,1\}^m} \langle Ax, (t + e_m) \rangle = \max_{x \in \{0,1\}^n, (t,\tau) \in \{-1,1\}^{m+1}} \langle Bx, (t, \tau) \rangle.$$

One can then construct a two layer neural network, where the first weight matrix is $B^T$, and the second weight matrix is $(1, \ldots, 1) \in \mathbb{R}^n$. Because the network we consider has only one output, the second weight matrix is only a vector. The FGL of this network is exactly twice of the cut norm of $A$. $\qquad\square$

## A.7 Proof of Theorem 3.3

*Proof.* Let $B = \begin{pmatrix} 0 & 0 & 0 \\ A & Ae_n & 0 \end{pmatrix}$. Combing Equations (5), (17) and (18), the approximation algorithm for Equation (4) where $q = 1$ is induced by the following SDP program:

$$\max \frac{1}{2}\mathrm{tr}(BX)$$
$$s.t.\ X \succeq 0, X_{ii} = 1, i \in [n+m+1].$$

$\square$

## A.8 Natural SDP relaxation of $\ell_2$-FGL estimation

Now let $q = 2$ in Equation (4), we will have:

$$\max_{y \in \{0,1\}^n} ||Ay||_2.$$

In other words, we only need to solve the following program:

$$\max_{z \in \{0,1\}^n} z^T(A^T A)z. \tag{19}$$

Let $M = A^T A$, then $M$ is a PSD matrix. We have demonstrated the scaling techniques in Appendix A.5. Let $x \in \{-1,1\}^{n+1}$, one can verify that

$$\max_{z \in \{0,1\}^n} z^T M z = \frac{1}{4} \max_{x \in \{-1,1\}^{n+1}} x^T \hat{M} x, \tag{20}$$

where $\hat{M} = \begin{pmatrix} M & Me_n \\ e_n^T M & e_n^T Me_n \end{pmatrix}$.

It is easy to verify that if $M$ is PSD, $\hat{M}$ is also PSD. Because $M = A^T A$, $\hat{M} = (A, Ae_n)^T \cdot (A, Ae_n)$. Now we can consider the following natural SDP relaxation to $\max_{x \in \{-1,1\}^{n+1}} x^T \hat{M} x$:

$$\max \mathrm{tr}(\hat{M}X)$$
$$s.t.\ X \succeq 0, X_{ii} = 1, i \in [n+1]. \tag{21}$$

This SDP relaxation admits a $\frac{\pi}{2}$-approximation factor from Equation (16) (Rietz, 1974; Nesterov, 1998).

## A.9 Proof of Theorem 3.4

*Proof.* Let $\hat{M} = \begin{pmatrix} M & Me_n \\ e_n^T M & e_n^T Me_n \end{pmatrix}$, where $M = A^T A$. Combining Equations (19) to (21), the approximation algorithm for Equation (4) where $q = 2$ is induced by the following SDP program:

$$\max \frac{1}{2}\sqrt{\mathrm{tr}(\hat{M}X)}$$
$$s.t.\ X \succeq 0, X_{ii} = 1, i \in [n+1]. \tag{22}$$

$\square$

## A.10 Comparison with Raghunathan et al. (2018)

Raghunathan et al. (2018) formulated the following SDP to upper bound the $\ell_\infty$-FGL on two-layer neural networks:

$$\max \frac{1}{4}\mathrm{tr}(CX)$$
$$s.t.\ X \succeq 0, X_{ii} = 1, i \in [n+m+1], \tag{23}$$

where $C$ is a $(m + n + 1) \times (m + n + 1)$ matrix, and $C = \begin{pmatrix} 0 & 0 & A^T \\ 0 & 0 & e_n^T A^T \\ A & Ae_n & 0 \end{pmatrix}$.

If we compare Equations (6) and (23), $C = B + B^T$. Because $X$ is symmetric, $\text{tr}(CX) = 2\text{tr}(BX)$. Therefore, Equations (6) and (23) produce the same result.

### A.11 Equivalence between the new optimization program and Equation (19)

Notice that because $u \in \mathbb{R}^{1 \times n}$, $u\Delta\sigma(x)$ is a scalar. We can view each $z_i$ in Equation (19) as $\frac{\Delta\sigma(x)_i}{\Delta y_i}$, the derivative of $\sigma(x)_i$ without the limit. Therefore, $\Delta\sigma(x)_i = z_i \Delta y_i$. Recall that from Section 4, $\Delta y_i = w_i \Delta x$, so $\Delta\sigma(x)_i = w_i z_i \Delta x$, then $u\Delta\sigma(x) = \Delta x \sum_i^n u_i z_i w_i = \Delta x (Az)$, where $A = W^T \text{diag}(u)$ as defined in Equation (4).

As a result, from Cauchy–Schwarz inequality, the above objective is

$$\max_{\Delta x, \Delta\sigma(x)} \frac{(u\Delta\sigma(x))^2}{(\Delta x)^2} = \max_z (Az)^2,$$
$$s.t. \ z \in [a, b]^n.$$

This demonstrates the equivalence between the new optimization program and Equation (19) when $[a, b] = [0, 1]$ for $\sigma = \text{ReLU}$.

## B  Polynomial optimization approach to the FGL estimation

We briefly discuss the gradient approach to estimate the FGL. Let us use a three layer network as an example:
$$f(x) = u\sigma(V\sigma(Wx + b_1) + b_2),$$
where $x \in \mathbb{R}^{l \times 1}$, $W \in \mathbb{R}^{n \times l}$, $b_1 \in \mathbb{R}^n$, $V \in \mathbb{R}^{m \times n}$, $b_2 \in \mathbb{R}^m$ and $u \in \mathbb{R}^{1 \times m}$.

The formal gradient vector of this network is
$$W^T \text{diag}(y) V^T \text{diag}(z) u^T,$$
where $\text{diag}(y) \in \mathbb{R}^{n \times n}$ and $\text{diag}(z) \in \mathbb{R}^{m \times m}$. The $i$-th component of this vector is then

$$\sum_{k=1}^m \sum_{j=1}^n (u_k V_{kj} W_{ji}) \cdot (y_j z_k).$$

Therefore, the $\ell_p$-norm estimation of the formal gradient ends up being a polynomial optimization problem over a cube. For example, the $\ell_1$-norm (corresponding to $\ell_\infty$-perturbations) of the gradient is

$$\max_{x_i \in \{-1,1\}, y_j \in \{0,1\}, z_k \in \{0,1\}} \sum_{i,j,k=1}^{l,n,m} T_{ijk} \cdot x_i y_j z_k, \tag{24}$$

where $T_{ijk} = W_{ji} V_{kj} u_k$.

This is essentially a tensor cut-norm problem, and it is an open problem whether there exists an approximation algorithm within a constant factor to the general tensor cut-norm problem (Kannan, 2010). Notice that Equation (24) is not a general tensor-cut-norm problem, because the tensor is generated from the weight matrices. For example, if we fix $j$, the projected matrices of $T$ are of rank-1. Each vector in $T_{:,j,:}$ is the product of $V_{kj} u_k$ with the vector $W_{j:}$:

$$\forall k : T_{:,j,k} = W_{j:} V_{kj} u_k.$$

However, we do not have the theoretical technique to exploit the low-rank structure of these special polynomial optimization problems. The perturbation analysis in Sections 4 and 5 can be viewed as exploiting this structure in practice.

Table 3: $\ell_\infty$-FGL estimations of different methods for two-layer networks: DGeoLIP and NGeoLIP induce the same estimations, and they are also close to the sampled lower bounds. In the meantime, the result from GeoLIP is tighter than LiPopt's result.

| #Units | DGeoLIP | NGeoLIP | LiPopt-2 | MP | Sample | BruF |
|---|---|---|---|---|---|---|
| 8 | 142.19 | 142.19 | 180.38 | 411.90 | 134.76 | 134.76 |
| 16 | 185.18 | 185.18 | 259.44 | 578.54 | 175.24 | 175.24 |
| 64 | 287.60 | 287.60 | 510.00 | 1207.70 | 253.89 | N/A |
| 128 | 346.27 | 346.27 | 780.46 | 2004.34 | 266.22 | N/A |
| 256 | 425.04 | 425.04 | 1011.65 | 2697.38 | 306.98 | N/A |

Table 4: Average running time (in seconds) of different methods for two-layer-network $\ell_\infty$-FGL estimations: GeoLIPs are faster than LiPopt.

| # Hidden Units | DGeoLIP | NGeoLIP | LiPopt-2 | BruF |
|---|---|---|---|---|
| 8 | 23.1 | 21.5 | 1533 | < 0.1 |
| 16 | 28.1 | 22.3 | 1572 | 4.8 |
| 64 | 93.4 | 31.7 | 1831 | N/A |
| 128 | 292.5 | 42.2 | 2055 | N/A |
| 256 | 976.0 | 70.9 | 2690 | N/A |

## C    Complete experimental specifications and results

GeoLIP is available at `https://github.com/z1w/GeoLIP`. To accommodate users who do not have access to MATLAB, we also implement a version based on CVXPY (Diamond and Boyd, 2016). However, the MATLAB implementation works more efficiently in terms of memory and speed, and we encourage users to work with the MATLAB version when possible. We conducted all the GeoLIP-related experiments with the MATLAB version.

### C.1    Experimental specifications

**Tools.** We obtain the LiPopt implementation from `https://github.com/latorrefabian/lipopt`, under the MIT License.

**Server specification.** All the experiments are run on a workstation with forty-eight Intel® Xeon® Silver 4214 CPUs running at 2.20GHz, and 258 GB of memory, and eight Nvidia GeForce RTX 2080 Ti GPUs. Each GPU has 4352 CUDA cores and 11 GB of GDDR6 memory.

**Dataset and split.** We used the standard MNIST dataset from the PyTorch package (Paszke et al., 2019). We used the "train" parameter in the MNIST function to split training and testing data.

### C.2    Experimental results

**Single hidden layer.** We consider the $\ell_\infty$-FGL estimation on two layer neural networks with different numbers of hidden units. The results are summarized in Tables 3 and 4.

**Multiple hidden layers.** We consider the $\ell_\infty$-FGL estimation on 3, 7, 8-layer neural networks. Each hidden layer in the network has 64 ReLU units. The results are summarized in Tables 5 and 6.

**$\ell_2$-FGL estimation.** We measure the $\ell_2$-FGL on two-layer networks mainly to compare whether Equation (7) and LipSDP produce the same result. Additionally, we also want to empirically examine the approximation guarantee from Theorem 3.4. Still, we consider networks with 8, 16, 64, 128, 256 hidden nodes. The results are summarized in Tables 7 and 8.

Table 5: $\ell_\infty$-FGL estimations of different methods for multi-layer networks: GeoLIP's result is much tighter than the matrix-product method, and LiPopt is unable to handle these networks.

| # Layers | GeoLIP | Matrix Product | Sample | LIPTOPT |
|---|---|---|---|---|
| 3 | 529.42 | 9023.65 | 311.88 | N/A |
| 7 | 5156.5 | $1.423 * 10^7$ | 1168.8 | N/A |
| 8 | 8327.2 | $8.237 * 10^7$ | 1130.6 | N/A |

Table 6: Average running time (in seconds) of GeoLIP for multi-layer-network $\ell_\infty$-FGL estimations.

| 3-Layer Net | 7-Layer Net | 8-Layer Net |
|---|---|---|
| 120.9 | 284.3 | 329.5 |

## C.3 Additional discussions

**Duality.** The results in Table 7 show that the results of LipSDP and GeoLIP on two-layer-network $\ell_2$-FGL estimation are exactly the same, which empirically demonstrates the duality between LipSDP and GeoLIP, as discussed in Section 4. Though Pauli et al. (2022) showed that the most precise version of LipSDP is invalid for estimating an upper bound of $\ell_2$-FGL, our dual-program argument shows that the less precise version of LipSDP is correct.

**Precision.** We showed that GeoLIP's approximation factor for the $\ell_2$-FGL estimation on two layer networks is $\sqrt{\frac{\pi}{2}} \approx 1.253$ in Theorem 3.4. The $\ell_2$-FGL from GeoLIP is very close to the sampled lower bound of true Lipschitz constant in Table 7. On the other hand, because the result from GeoLIP is an upper bound of FGL, and this result is not much greater than the sampled lower bound of true Lipschitz constant, this empirically demonstrates that the true Lipschitz constant is not very different from the FGL on two-layer networks.

**Running time.** If we compare the running time in Tables 4 and 8, the dual program takes more time to solve than the natural relaxation. This is particularly true when the number of hidden neurons increases. From the reported numbers of variables and equality constraints by CVX, the dual program and natural relaxation have similar numbers. It is also observed that the CPU usage is higher when the natural relaxation is being solved. We want to point out that the running time and optimization algorithm are solver-dependent, and efficiently solving SDP is beyond the scope of this work. It is an interesting future direction to exploit the block structure of the dual programs, and develop algorithms that are compatible with those programs, because training smooth networks is a critical task, and it is promising to incorporate the SDP programs.

**$\ell_2$ versus $\ell_\infty$ FGLs.** If we compare results from Tables 3 and 7, we can also find that the discrepancy between matrix product method and sampled lower bound is much smaller in the $\ell_2$ case. This could also explain why Gloro works for $\ell_2$-perturbations but not the $\ell_\infty$ case in practice, where Leino et al. (2021) used matrix-norm product to upper bound the Lipschitz constant of the network in Gloro.

**Sampling.** Sampling can only give a lower bound of the true Lipschitz constant, while we are trying to estimate an upper bound. We use sampling as a sanity check to ensure that the SDP method is at least sound and indeed provides an upper bound of the FGL. It is interesting to see that in networks where we can brute-force enumerate all the activation patterns, sampling provides very close results to the ground-truth ones. Notice that for those networks, there are only a few hidden units (8 or 16), while we sample many (200,000) inputs, which might activate all or most of the patterns. However, for networks with many activation nodes, it is infeasible to have a brute-force enumeration of all the activation patterns, so we do not have the ground-truth information. Sampling has no guarantee whether it can activate all patterns unless we have sampled all possible inputs, which is also impractical.

**Multi-layer network guarantees.** The discrepancy between the results from sampling and GeoLIP is relatively large for multi-layer networks. The approximation guarantee of GeoLIP is in terms of

Table 7: $\ell_2$-FGL estimations of different methods for two-layer networks: LipSDP and NGeoLIP induce the same estimations, and these results are also close to the sampled lower bounds.

| #UNITS | NGEOLIP | LIPSDP | MP | SAMPLE | BRUF |
|---|---|---|---|---|---|
| 8 | 6.531 | 6.531 | 11.035 | 6.527 | 6.527 |
| 16 | 8.801 | 8.801 | 13.936 | 8.795 | 8.799 |
| 64 | 12.573 | 12.573 | 22.501 | 11.901 | N/A |
| 128 | 15.205 | 15.205 | 30.972 | 13.030 | N/A |
| 256 | 18.590 | 18.590 | 35.716 | 14.610 | N/A |

Table 8: Average running time (in seconds) of LipSDP and NGeoLIP for two-layer-network $\ell_2$-FGL estimations.

| # HIDDEN UNITS | LIPSDP | NGEOLIP | BRUF |
|---|---|---|---|
| 8 | 11.5 | 1.2 | < 0.1 |
| 16 | 15.7 | 1.2 | 5.1 |
| 64 | 64.2 | 1.3 | N/A |
| 128 | 216.1 | 1.7 | N/A |
| 256 | 758.1 | 4.1 | N/A |

the FGL, rather than true Lipschitz constant. It is unclear how large the gap between true Lipschitz constant and the FGL is for multi-layer networks. Narrowing this gap is an interesting research direction and beyond the scope of this work. We do not know whether for multi-layer networks, GeoLIP has an approximation guarantee that is independent of the network. We leave this as an open problem.

## D   Negative societal impacts

Our work is mainly theoretical and to measure an intrinsic mathematical property of neural networks, and can benefit the verification of deep-learning systems. A misuse of our work can give a false sense of safety, so the practical use of our work should be careful.