# OpenReview forum: "A Quantitative Geometric Approach to Neural-Network Smoothness"
_NeurIPS.cc/2022/Conference — NeurIPS 2022 Accept_

### Official Review · Reviewer_xmVx · 2022-07-03

**Rating:** 5
**Confidence:** 3
**Soundness:** 2 fair
**Presentation:** 1 poor
**Contribution:** 1 poor

**Summary:**

The paper deals with estimating the Lipschitz constants of neural networks, aiming to unify different approaches based on semidefinite programming (SDP). More precisely, the authors would like to estimate the "formal global Lipschitz constant" (FGL) which is an upper-bound of the actual Lipschitz constant that's however sharper than the naive product of the norms of all weight matrices. They consider FGL estimation for two layer networks in the $\ell_\infty$ and $\ell_2$ topologies and also outline FGP estimation for multi layer networks. Furthermore, they show that some existing SDP based models are duals of the FGP problem. In their numerical results section the authors compare their formulations to some other methods which estimate the Lipschitz constant, like dual SDP problems or brute force methods.

########POST-REBUTTAL#########

Since there seems to be relatively strong support for the paper, I increase my score to 5 and ask the authors improve the organization of the paper.

**Questions:**

I have a couple of questions and remarks that should be addressed:

- p.3, l.103: I think the $\operatorname{diag}$ operator shouldn't be part of the forward pass of the network. Is this a typo?
- p.3, l.112, eq. (3): I guess that the matrices $W^i$ for $i=2,\dots,d-1$ should also appear in this formula, right?
- p.3, l.122: an interval of the form $[a,b]$ can never coincide with the set $\lbrace 0,1\rbrace$.
- p.3, l.131: I don't understand why $f(x)$ has this form. This is linear in $x$. Also: what's $y$ in $\operatorname{diag}(y)$?
- p.7, l.242: same thing here
- In general, the presentation of sections 3 and 4 is suboptimal and lacks a golden thread.
- The numerical results in Table 1 seem to suggest that sampling is the best way to estimate the Lipschitz constant. Maybe you should compare your methods with sampling based Lipschitz constant estimation methods like for instance [2].
- Given that Lipschitz regularization of neural networks is a very active field, the related work section and introduction does not paint a clear picture of the different approaches out there, such as e.g. the articles [1-5].

[1] Aziznejad, S., Gupta, H., Campos, J., & Unser, M. (2020). Deep neural networks with trainable activations and controlled Lipschitz constant. IEEE Transactions on Signal Processing, 68, 4688-4699.

[2] Bungert, L., Raab, R., Roith, T., Schwinn, L. and Tenbrinck, D., 2021, May. CLIP: Cheap Lipschitz training of neural networks. In International Conference on Scale Space and Variational Methods in Computer Vision (pp. 307-319). Springer, Cham.

[3] Gouk, H., Frank, E., Pfahringer, B., & Cree, M. J. (2021). Regularisation of neural networks by enforcing lipschitz continuity. Machine Learning, 110(2), 393-416.

[4] Krishnan, V., Makdah, A., AlRahman, A., & Pasqualetti, F. (2020). Lipschitz bounds and provably robust training by Laplacian smoothing. Advances in Neural Information Processing Systems, 33, 10924-10935.

[5] Terjék, D. (2019). Adversarial lipschitz regularization. arXiv preprint arXiv:1907.05681.





**Limitations:**

To my mind, a negative societal impact cannot be expected and is also not discussed. Also there is a discussion section, the possible shortcomings (and advantages) of the proposed method over sampling type methods for estimating the Lipschitz constant is not discussed.

**Strengths And Weaknesses:**

Strengths: The numerical results seem to indicate that the presented method computes the same estimate for the Lipschitz constant as its dual SDP variants, however, in shorter time. Furthermore, it compares favorably against the chosen baseline method LiPopt and brute force estimatation.

Weaknesses: I find the paper pretty poorly written and it's not clear what exactly the contribution or the novelty is. The theoretical part of the paper is relatively ad hoc. Furthermore, the numerical results are not entirely conclusive, see "Questions" further down.

---

> ### Author Response · Authors · 2022-07-31
> **Responses to questions and concerns**
>
> We appreciate your review. Here are our response to the concerns:
>
> - In the general response section, we provide a brief explanation and overall contribution of this work. We hope it can help clarify some of the concerns on weakness.
>
> - For questions related to the forward network notations. We apologize for the notational confusion and have already updated the paper. See **C1.Notational confusion for forward networks** in the general response section for more explanation.
>
> - [a, b] versus {0, 1}: [a, b] is used to denote the derivative range of the activation function, and {0, 1} are the (almost everywhere) derivative values of ReLU. We have changed the presentation on {0, 1} to [0, 1] to avoid unnecessary technical details in the main paper, and provide more explanation why we can use {0, 1} in Appendix A1. See **C6. Intervals or vertices on the hypercube** in the general response section for more discussion.
>
> - Sampling seems the best way to estimate the Lipschitz constant: sampling can only provide a lower bound of the true Lipschitz constant, and it is supposed to be lower than the upper bound computed in our experiments, so this is used as a sanity check. See **C5. Sampling** in the general response section for more discussion.
>
> - Related works on Lipschitz regularization: Lipschitz regularization and Lipschitz measurement of neural networks are related, but different problems and subjects. We will add a paragraph of Lipschitz regularization in the paper and cite those works once we are allowed more space in the main paper.
>
> - Negative societal impact: we discussed this in the checklist, and add a section in the updated appendix.
>
> We are happy to address any further concerns and questions.

---

> > ### Comment · Reviewer_YiVe · 2022-08-07
> > **Sampling is not relevant here**
> >
> > As a fellow reviewer, I'd like to add that I agree with authors regarding the limitations pointed out by reviewer xmVx.
> >
> > Sampling can only yield lower bounds. So, it is fact complementary with approaches based on SDP relaxations.
> >
> > It does not make sense to compare sampling and SDP. They need to be used in conjonction to "sandwich" the true value of the Lipschitz constant.
> >
> > Hence, the paper is a good contribution on its own that does not require comparison with sampling. The closeliness between true Lipschitz constant, sampling, and GeoLip bound is clearly in favor of the paper.

---

> > > ### Comment · Reviewer_xmVx · 2022-08-08
> > > **Sampling**
> > >
> > > Thanks for pointing this out. A missing discussion or comparison with sampling methods is not my main criticism of the paper but rather its relatively poor presentation. I just got struck by the fact that the sampled approximations of the Lipschitz constant precisely agree with the brute force estimations whereas the GeoLIP method yields a conservative upper bound in these cases.

---

> > ### Comment · Reviewer_xmVx · 2022-08-08
> > **Thanks for the response**
> >
> > Thanks for your response and the clarifications.
> >
> > Regarding the sampling issue: I am well aware that this only provides a lower bound and that smaller values in the tables don't indicate better performance. Nevertheless, I was surprised to see that sampling gives precisely the same values as the brute force estimation in table s 1 and 3 whereas GeoLIP and the other approaches give strictly larger upper bounds.
> >
> > I would appreciate if the revised version of the paper would have an improved structure and explanations as clear as in the general response of the authors.

---

> > > ### Author Response · Authors · 2022-08-08
> > > **Thank you for your continuing effort**
> > >
> > > We appreciate your continuing engagement in the discussion.
> > >
> > > Regarding the measurement of sampling, it is possible that the network which we can brute-force search has only a few activation nodes (8 or16), and sampling many (200,000) inputs can activate all or most of the patterns. However, for many activation nodes, it is infeasible to have a brute-force enumeration of all the activation patterns, so we do not have groud-truth information. Sampling has no guarantee whether it can activate all patterns unless we have sampled all possible inputs, which is also impractical. We will add a discussion on this observation.
> > >
> > > All reviewers have provided presentational suggestions, and we have incorporated them into our updated paper. We appreciate all the advice and are more than happy to take further suggestions that can improve our paper.
> > >
> > > We believe that our work has lots of theoretical insights and practical implications (which is also hard to include in a paper with a small page budget), especially on why we can unify the $\ell_2$ and $\ell_\infty$-perturbations. On Lipschitz constant (upper bound) estimations, we do not know any previous works that can achieve this. We hope that our approach, in particular, the application of tools from quantitative geometry that characterize the relationship between metrics, can bring awareness to the relationship between metric geometry and deep learning, and bring more ideas to the investigation of neural networks.

---

### Official Review · Reviewer_YiVe · 2022-07-10

**Rating:** 8
**Confidence:** 2
**Soundness:** 4 excellent
**Presentation:** 3 good
**Contribution:** 4 excellent

**Summary:**

In this paper the authors propose a new tool to estimate an upper bound on the Lipschitz constant of ReLU based feed forward networks. They focus on Lipschitz constant for L-infinity norms and L2 norms. The computation of the upper bound is based on SDP relaxations of the original problem, a standard approach of the literature.

For L-infinity, as said by the authors, they "provide a rigorous derivation and simpler formulation, and also a sound theoretical analysis of the bound, which illustrate more insights to this problem" compared to the work of Raghunathan et al. (2018). Whereas for for L2-bounds they " show that LipSDP is dual of Equation (7) to estimate the ℓ2-FGL on two-layer networks".

Finally the release a software, GeoLip, that implements the SDP of interest for multil-layers neural networks (for l-infinity) and 2-layers neural networks (for l2). Not only the upper bound is improved compared to concurrent methods, but it is even cloaser to the lower bound obtained by sample or brute force by a small factor, even on deep networks. Moreover the running time is competitive even on those deep networks.

**Questions:**

l 125 : "However, the algorithms presented in this work can be adapted with minor adjustments to other common activation functions."
Does it also work with non elementwise activation functions ? In this case the diag(y) is no longer diagonal. Is it a simple way to adapt the method to arbitrary diag(y) ?

From my limited understanding of the topic, I have two more naive (related) questions.

* For multi-layers ( i.e > 2) neural networks , do you have any theoretical guarantees like the ones of theorems 3.3 and 3.4 ?
* experiments on l2 norms for multi-layers neural networks are lacking. Is it because the duality result of section 4.1 does not extend further than the case of two-layers networks ?

**Limitations:**

Every result is clearly stated along with its hypothesis. The only part that could be subject to caution is the the running time of their solver, which might unfairly benefit from better software than other SDP based relaxations of literature, as acknowledged by the authors: "Notice that the running time is implementation and solver-dependent".

**Strengths And Weaknesses:**

Strengths:

The paper makes a clear theoretial contribution by posing rigorously posing the problem. The link the Grothendieck inequality is very interesting and establish a clear bound toward algorithms must tend. The link between dual formulation (7) and previous work LipSDP
is interesting. Claims in all theorems 3.2 3.3 and 3.4 seem to be significant contributions.

The SDP are clearly specified which allows independent implementation, and easier reproducibility.

The speed and quality of the bounds of GeoLip software, whose code is made public, is very convincing and is validating theoretical claims.

The paper is well written, I learned a lot about this topic with which I am unfamiliar.

Weaknesses:

for reader unfamiliar with the topic some notations are a bit harsh: it took me some time to understand the motivation behind the definition of two-layers networks (l 135) : no bias since it does not play any role in computation of the upper bound (only in the pattern of activations between true Lipschitz constant and the FGL). diag(y) the pattern of activations, and u the weight of the last layer (with the NN seen as a function from R^n to R)

minor/typo: a parenthesis is lacking in equation (2)

---

> ### Author Response · Authors · 2022-07-31
> **Responses to questions and concerns**
>
> We appreciate your careful review and are excited that you found our paper helpful. Here are our responses to the questions and concerns:
>
> - We have updated the paper with respect to the forward network notations. Hopefully, this improves the presentational weakness. See **C1.Notational confusion for forward networks**  in the general response section for more clarification. We also provide a brief overview of our work in the general response section, which might be helpful to clarify the weakness concern.
>
> - Adaptation to non-elementwise activation functions: In the most general sense, we only need to write the problem of interest as a quadratic program, and then we can apply Shor's relaxation to obtain an SDP. See **C2. Applicability of the SDPs** in the general response section for more discussion. Whether we can extend the (dual) SDP to non-elementwise activations should depend on what the activation is, and whether we can use quadratic constraints to encode the activation. We are more than happy to discuss concrete examples if there are any.
>
> - Multi-layer theoretical guarantees: We do not know theoretical guarantees for multi-layer networks, even though our dual interpretation is the precise SDP relaxation (in the dual sense) of the FGL estimation, and provides a practical algorithm. However, it is unclear how the dual interpretation can provide a theoretical guarantee, and we leave this as an open problem. See **C4. Multi-layer network theoretical guarantee** in the general response section for more discussion.
>
> - Multi-layer $\ell_2$-norm experiments: you are right because we do not have a duality program for multi-layer $\ell_2$-FGL estimation; also there are few benchmark tools for $\ell_2$-FGL estimation after LipSDP was known. The experiment on $\ell_2$-FGL for multi-layer networks would be comparing LipSDP-neuron with sampling and matrix-norm product, and we should not take credit for LipSDP-neuron's empirical excellence.

---

> > ### Comment · Reviewer_YiVe · 2022-08-07
> > **Thanks for the precisions**
> >
> > Thank you for your precisions.
> >
> > I was curious about **GroupSort** activation function. It is defined as follow :
> >
> > $\text{GroupSort}(x_1,x_2)=(\max(x_1,x_2),\min(x_1,x_2))$
> >
> > Like ReLU it is piecewise affine, but operate on **pairs** of consecutive neurons.
> >
> > It is more expressive than ReLU. See the paper below:
> >
> > Tanielian, U. and Biau, G., 2021, March. Approximating Lipschitz continuous functions with GroupSort neural networks. In International Conference on Artificial Intelligence and Statistics (pp. 442-450). PMLR.
> >
> > It is used to parametrized 1-Lipschitz NN since the Jacobian is orthogonal. It avoids vanishing gradients and benefit from universal approximation theorems.
> >
> > Do you believe a quadratic program can encode those activations ?

---

> > > ### Author Response · Authors · 2022-08-08
> > > **Thanks for your continuing effort**
> > >
> > > Thanks for your continuing effort in our discussion. If our understanding of GroupSort activations is correct, we can encode GroupSort activations in quadratic relations.
> > >
> > > Because GroupSort activates on pairs of consecutive neurons, and the semantics of group sort is to either preserve a fixed pair or switch the pair, the FGL is a maximization problem over all possible switching or keeping the pairs.
> > >
> > > Let $\Delta x$ and $\Delta y$ denote the input perturbations on a pair of neurons, and $\Delta u$ and $\Delta v$ be the output perturbation of the pair of GroupSort neurons. The relations we want to encode is $\Delta u = \Delta x$ and $\Delta v = \Delta y$, or $\Delta u = \Delta y$ and $\Delta v = \Delta x$.
> > >
> > > 1. It is not hard to encode $\Delta u = \Delta x$ or $\Delta u = \Delta y$: $(\Delta u - \Delta x)(\Delta u - \Delta y)=0$.
> > >
> > > 2. Similarly, we can have $(\Delta v - \Delta x)(\Delta v - \Delta y)=0$ for $\Delta v = \Delta x$ or $\Delta v = \Delta y$.
> > >
> > > Then we need to make sure $\Delta u = \Delta x$ and $\Delta v = \Delta y$, or $\Delta u = \Delta y$ and $\Delta v = \Delta x$, not other cases (for example, $\Delta u = \Delta x$ and $\Delta v = \Delta x$ also satisfies the two constraints above). Notice that if $\Delta x = \Delta y$, then we are already done. We need to consider when $\Delta x \neq \Delta y$. We can add an extra constraint: $(\Delta u - \frac{\Delta x+\Delta y}{2})(\Delta v - \frac{\Delta x+\Delta y}{2})\leq 0$. This constraint would make sure that $\Delta u \neq \Delta v$ given $(\Delta v - \Delta x)(\Delta v - \Delta y)=0$ and $(\Delta u - \Delta x)(\Delta u - \Delta y)=0$.
> > >
> > > Notice that to apply Shor's relaxation [Chapter 4.3.1, 1], we need inequality relations, so $a=b$ will be two constraints: $a\geq b$ and $a\leq b$. As a result, we need five constraints for each pair of neurons, and then introduce five dual variables in Shor's relaxation, instead of one dual variable for one neuron in the elementwise activation cases.
> > >
> > > For the elementwise quadratic encoding, we would have $(\Delta u-a\Delta x)(\Delta u- b\Delta x)\leq 0$, corresponding to $a \leq\frac{\Delta u}{\Delta x}\leq b$ (in the ReLU case, $b=1$ and $a=0$). For pairwise neurons, $\Delta u$, $\Delta v$, $\Delta x$ and $\Delta y$ are entangled. $\Delta x$ and $\Delta y$ are still the linear transformations from the previous layer output perturbations as in the elementwise activation network case; i.e., $\Delta x = w\cdot \Delta z$, where $w$ is the weight vector corresponding to the neuron, and $\Delta z$ denote the output perturbation from the previous layer.
> > >
> > > [1] Aharon Ben-Tal and Arkadi Nemirovski. 2001. Lectures on Modern Convex Optimization. Society for Industrial and Applied Mathematics. https://doi.org/10.1137/1.9780898718829

---

### Official Review · Reviewer_xMmc · 2022-07-11

**Rating:** 5
**Confidence:** 2
**Soundness:** 3 good
**Presentation:** 2 fair
**Contribution:** 3 good

**Summary:**

This paper proposed a new method for estimating the Lipschitz constant (more specifically, the ($\ell^p$-) Formal Global Lipschitz (FGL) constant) of DNNs.
First, this paper analyzed two-layer ReLU-DNNs and showed that $\ell^{\infty}$-FGL estimation is MAXSNP-hard (Theorem 3.2). Then, by relaxing the FGL-estimation problem and reducing it to SDP, this paper derived a polynomial algorithm with an approximation ratio $K_G$ (Theorem 3.3) for general $p=\infty$ and $\sqrt{\frac{\pi}{2}}$ for $p=2$ (Theorem 3.4).
Next, the $\ell^p$-FGL estimation problem of two-layer NNs was formulated as the maximization of perturbation of intermediate units with respect to that of the input. Then, this paper showed that this formulation is equivalent to the first formulation (l.211) and is dual to LipSDP (l.212--220) when $p=2$. In addition, this paper extended it to multi-layer NNs.
Finally, this paper evaluated the proposed method numerically and claimed that the proposed method performed tighter estimation than existing methods for the $\ell^{\infty}$-estimation problem of two-layer and multi-layer DNNs.

**Questions:**

- l.103: $f_i(x)=W^i\mathrm{diag}(\sigma(f_{i-1}(x))) + b_i$ → I think $\mathrm{diag}$ is not needed here.
- l.107: We do not usually use $\prod$ for matrix multiplications, which are non-commutative.
- l.160: Although we can understand the meaning of the term "approximation ratio" intuitively, I would suggest writing its mathematical definition explicitly since it is used in mathematical statements.
- l.169 (7): $A\cdot B$ is undefined.
- l.209: $f(x) = u\sigma(x)$ → $f(x) = u\sigma(Wx)$ ?
- l.563: I was not aware that $x$ in l.138 takes value in {-1, 1}^n. I suggest writing the range of $x$ more explicitly. I think l.563 does not hold in general (at least without any assumption). We should have:

\max_{x\in \{-1, 1\}^n} \|Ax\|_{1} = \max_{x\in \{-1, 1\}^n, , y\in \mathbb{R}^m, \|y\|_{\infty}=1} \langle Ax, y\rangle

So, I am wondering why we can restrict the range of $y$ to {-1, 1}^{m}

**Limitations:**

As far as I have checked, this paper did not discuss the proposed methods' limitations. However, the proposed method (1) only applies to ReLU-DNNs and (2) has no theoretical guarantees for multi-layer NNs. Therefore, it is desirable to discuss such limitations.

**Strengths And Weaknesses:**

Strengths

- NGeoLip proposed in this paper has theoretical guarantees for the FGL-estimation problem of two-layer NNs.
- DGeoLIP proposed in this paper applies to multi-layer NNs, while competitive methods are limited to two-layer NNs.
- SDP allows a unified approach to $\ell^2$-FGL and $\ell^\infty$-FGL.

Weaknesses

- This paper is intended to apply the Lipshitz constant estimation to mitigate the vulnerability against adversarial attacks. However, this paper did not experimentally evaluate the effectiveness of the proposed methods in adversarial attacks.
- There is room for discussion on whether the proposed method sufficiently demonstrates its effectiveness (either theoretically or empirically) to the estimation problem in the multi-layer settings.
- There is room for improvements in the organization of the paper (see Clarity section).


Originality（Novelty）

- As this paper mentioned, existing literature employed the idea of reducing the Lipschitz constant estimation problem to SDP (l.55). However, this paper adopted this idea differently and proposed different algorithms.
- The authors claimed their work is guided by the principle that we should separate geometry-dependent and independent components. The authors think that this is a crucial idea underlying this paper. However, it was not clear to me how this principle works. For example, in Sections 3.1 and 3.2, $\ell^\infty$-FGL and $\ell^2$-FGL were first reduced to the mixed-norm problem and then were analyzed separately. In this example, is it correct to understand that the reduction to the mixed-norm problem is the geometrically-independent part? I suggest the authors write how they applied this principle to actual algorithms.


Quality（Soundness）

- As far as I have checked, the mathematical statements and proofs are correct all in all.
- The details of the experimental setup are described in the Appendix. I can confirm the validity of the experiment.
- l.144: On which does the constant $K_G$ depend? Does $K_G$ depend on the Hilbert space $H$?


Clarity（Presentation）

- There is room for improvement in the organization of the paper. However, I think it does not significantly impact the paper's evaluation because I expect the authors can correct it quickly.
- I found Section 3.1 to be a bit difficult to read. Specifically, this section (1) explained the SDP relaxation for the matrix-norma problem (l.140--148), (2) changed the topic to the problem without relaxation and showed its hardness (l.149--152), and (3) went back to the problem with relaxation (l.153--169). Therefore, I suggest reconsidering the order of the three.
- I think the content of Section 4.1 should be in Section 3 rather than Section 4 since it is about two-layer NNs. Raghunathan et al. (2018), which applied two-layer NNs, is compared to eq. (6) in Section 3.1. Also, LipSDP and LipSDP-network appeared in Section 4.1. is compared with eq. (7) in Section 3.2.

Significance

- For two-layer NNs, this paper showed the theoretical and empirical effectiveness of the proposed methods well.
- For multi-layer NNs, the DGeoLIP does not have theoretical guarantees. In addition, it is not better than the baseline method by sampling (Table 1). Therefore, I have a question about the effectiveness of the proposed method for multi-layer settings.
- This paper intends to apply the estimation of the Lipschitz constant to mitigate the vulnerability of NNs to adversarial attacks. However, the numerical experiments are only performed on the pure Lipschitz constant estimation problem. Therefore, this paper would be more significant if it could show that the proposed method is effective against adversarial attacks in some way.

---

> ### Author Response · Authors · 2022-08-01
> **Responses to questions and concerns**
>
> We appreciate your detailed and rigorous review a lot. Here are our responses to the questions and concerns:
>
> - l103: We removed it.
>
> - l107: We expanded the multiplication in the equation.
>
> - l160: We added a definition in Appendix A.2.
>
> - l169 eq (7): We removed $\cdot$.
>
> - l209: We modified the notation to avoid confusion. Now we have $f(x) = u\sigma(y)$.
>
> - l563: This holds because for $x\in \mathbb{R}^n$, ||x||_1= |x_1|+...+|x_n| = \max_{y\in{-1, 1}^n}\langle x, y \rangle. See **C6. Intervals or vertices on the hypercube** in the general response section and Appendix A.1 paragraph *Maximum over hypercube* in the updated paper for more clarification.
>
> - Limitation 1: the proposed method does not only apply to ReLU-DNNs. The essential reason is that our quadratic constraint is compositional, and each of the constraints encodes a neuron computation. See **C2. Applicability of the SDPs** for more discussion.
>
> - Limitation 2: No theoretical guarantees for multi-layer networks. We do not know theoretical guarantees for multi-layer networks, though the dual program exploits the low-rank structure of the FGL-estimation problem. It is not clear to us how this implies a theoretical guarantee. We leave it as an open problem. **C4. Multi-layer network theoretical guarantee** provides more discussion.
>
> - Baseline method by sampling: sampling is expected to produce a lower value, and we used it as a sanity check, especially because LipSDP was shown to fail to produce an upper bound [1], and we want our upper bound to be at least sound. See **C5. Sampling** for more clarification.
>
> - Organizations: we reorganized section 3.1, and plan to isolate section 4.1 into a single section when more space is allowed.
>
> - The Grothendieck constant: $K_G$ is independent of the Hilbert space $H$ and the matrix $A$. That is why the Grothendieck inequality implies a universal $K_G$-approximation guarantee. If we impose additional assumptions on $A$ or $H$, the approximation ratio can be smaller.
>
> - The quantitative geometric principle: The SDP relaxation presented in section 3 is the precise SDP relaxation of a nonconvex optimization problem, and the approximation guarantee comes from the underlying geometric inequalities. As for the principle, we interpret LipSDP-neuron for $\ell_2$-FGL estimation as a relaxed compositional quadratic program. To transfer the techniques to the $\ell_\infty$ perturbations, we only changed the perturbation geometry encoding. In contrast, [2, 3] scaled LipSDP's result by $\sqrt{d}$ when transferring LipSDP's result to the $\ell_\infty$ setting, which we believe was a wrong transfer. See **C3. How is quantitative geometry related** for more discussion.
>
> - Evaluation in the context of adversarial robustness: Lipschitz continuity, an essential mathematical property of a function, plays an important role on many topics, including adversarial robustness and learning theory. Measuring the Lipschitz constant is a self-contained problem. For example, [3,4] did not evaluate the Lipschitz measurement in the context of adversarial robustness.  The evaluation of [2,5] in terms of adversarial robustness was to measure the Lipschitz constant of adversarially trained networks, so we conducted a similar experiment to measure the Lipschitz constant of PGD-adversarially trained networks. Selected results are presented below, and the code has been pushed to the repository.
>
> |   Network   | DGeoLIP | NGeoLIP | LiPopt | MP | Sample |
> | ----------- | ----------- | ----------| ----------| -----------| -------|
> |2-layer ,128 units, normally trained|    361.75    | 361.75 | 741.23 | 2049.77 | 294.66
> | 2-layer, 128 units, adversarially trained   | 54.49 |54.49 | 133.52 | 419.11 | 39.12
>
> |   Network   | DGeoLIP |MP | Sample |
> | ----------- | ----------- | ----------| ----------|
> |7-layer, 64 units per hidden layer, normally trained| 3782.94   |$1.123 * 10^7$ | 924.59
> |7-layer, 64 units per hidden layer, adversarially trained  |  424.68  |  $2.598 * 10^6$ | 58.85
>
> It is easy to see that PGD-adversarial training strongly regularizes the network Lipschitzness.
>
> [1]Patricia Pauli, Anne Koch, Julian Berberich, Paul Kohler, and Frank Allgöwer. 2022. Training Robust Neural Networks Using Lipschitz Bounds. IEEE 31 Control Systems Letters 6 (2022), 121–126.
>
> [2]Matt Jordan and Alexandros G Dimakis. 2020. Exactly Computing the Local Lipschitz Constant of ReLU Networks. NeurIPS 2020
>
> [3]Fabian Latorre, Paul Rolland, and Volkan Cevher. 2020.  Lipschitz constant estimation of Neural Networks via sparse polynomial optimization. ICLR 2020.
>
> [4]Kevin Scaman and Aladin Virmaux. 2018. Lipschitz Regularity of Deep Neural Networks: Analysis and Efficient Estimation. NIPS’18.
>
> [5]Mahyar Fazlyab, Alexander Robey, Hamed Hassani, Manfred Morari, and George Pappas. 2019. Efficient and Accurate Estimation of Lipschitz Constants for Deep Neural Networks. NeurIPS 2019

---

> > ### Comment · Reviewer_xMmc · 2022-08-08
> > **Post-rebuttal comments**
> >
> > I thank the authors for considering my comments and answering my questions seriously.
> >
> > - Questions: OK
> > - Limitation 1: I understand that the proposed methods apply activation functions with bounded derivatives and are not restricted to ReLU.
> > - Limitation 2: I understand that theoretical guarantees for multi-layer NNs are an open problem and therefore, their non-existence is not a big negative point.
> > - Comparison with sampling: I understand we cannot directly compare the proposed method with the sampling because it just gives a lower bound of the Lipschitz constant.
> > - Organization: OK
> > - Grothendieck constant: OK
> > - Qualitative geometric principle: I thank the authors for the explanation. If my understanding is correct, the authors claim that we can treat the Lipschitz constant estimation problems for different $p$ by changing the underlying geometry.
> > - Evaluation in the context of adversarial robustness: I appreciate the authors providing additional experiment results. My original intention was to evaluate how the proposed methods have a good effect on the performance of target NNs as machine learning models (e.g., prediction, estimation, generalization). I am sorry for the confusion. I understand that the proposed method gives a good Lipschitz constant estimation in the adversarial training setting.
> >
> > Additional question: Section 4.2 extends the dual SDP problem to the multi-layer NNs for $p=\infty$. Is it possible to do a similar extension for the $p=2$ setting?

---

> > > ### Author Response · Authors · 2022-08-08
> > > **Thanks for your continuing effort**
> > >
> > > We appreciate your continuing engagement in the discussion.
> > > - Changing the underlying geometry: yes.
> > >
> > > - Evaluation in the context of adversarial robustness: Thanks for the clarification. Given the limited time remaining for the discussion, we would not be able to provide additional experimental results. For example, we will need to compute 45 (10 choose 2) pairwise margin Lipschitz constants for each MNIST network, which is time-consuming to solve for the tools we are considering. Our existing experimental evaluation provides strong evidence to support the claims made in our work and is easy to reproduce.
> > >
> > > - Additional question: For $p=2$, the dual extension will give us exactly LipSDP-neuron. Because we give a compositional quadratic program interpretation of LipSDP-neuron, we can extend the structure in LipSDP-neuron to many new settings, and encode new computation structures beyond DNN. In contrast, LipSDP proposed an SDP as a whole, known as LipSDP-network, and then devised LipSDP-neuron as a simplified variant of LipSDP-network. Recently LipSDP-network was shown to fail to produce an upper bound of the Lipschitz constant, which would be an intuitive result under our framework. We will clarify this fact in Remark 4.3.
> > >
> > > We are happy to address any further concerns and questions. If our responses have addressed all the concerns and the revised paper has improved the merit of our work, we would appreciate an increased rating.

---

### Official Review · Reviewer_TC4r · 2022-07-12

**Rating:** 7
**Confidence:** 4
**Soundness:** 3 good
**Presentation:** 3 good
**Contribution:** 3 good

**Summary:**

The article tackle the estimation of the Lipschitz constant of a neural network for norms $p \in \{ 2, \infty \}$. The authors rely on previously existing SDP techniques and propose new formulations that give new theoretical insights on Lipschitz estimations.
Indeed, the authors bound the approximation of the polynomial SDP relaxation, and provide formulation for norms $2$ and $\infty$.
The theory is developed for NN with one hidden layer and then extend to the general case, giving a new algorithm called GeoLip.
Finally, the paper provide experimental results that shows that GeoLip outperforms in accuracy and time previous state of the art methods on multi-layers straightforward neural networks.

**Questions:**

Where does the name 'GeoLIP' comes from?

In the experimental section, are NGeoLIP results taking into account the approximation bound in order to be upper-bounds?

It would be interesting to show an example in which NGeoLIP and DGeoLIP give different bounds in order to better assess the advantage of using a (fast) approximation algorithm.

Would it be possible to extend the current approach to "much bigger" feed-forward network such as CNN or residual nets? It seems it is a difficult question as is, but would specific SDP solving strategies exploit the characteristics of these operations instead of their matrix form (which would make it intractable)?

**Limitations:**

Yes

**Strengths And Weaknesses:**

The paper is clear, well written and properly motivated. The other approach in the literature are well discussed and the paper is honest at comparing itself. Overall, I really enjoyed reading this paper.

In all generality, the dual of $\ell^\infty$ is ill-defined (l120) and requires some more hypothesis, even though it does not matter here.

l131:
    In the problem description describe what is $y$ (one line it is $\sigma$, the next one it is $\sigma'$ if I get it) and explain the simplification made from the definition of neural network above (no bias).
    Use notations defined before, what is y here?
    Could we write f(x) = W2 sigma (W1 x), which would then reduce to eq(4) directly thanks to eq(3), while preserving the notations introduced before.
l147: ref for the majoration of $K_G$ and its link to eq (5).


* Conventions

Globally, why upper indices? In the paper it seems that lower indices should be enough and this would make the paper even clearer.

* Typos:
    - l92: "X [is] positive semidefinite";
    - l88,92,146: Sentences beginning with a symbol;
    - l103: no diag in the equation;
    - l109: almost everywhere (no parenthesis);
    - l112: say.

Check the end of sentences in particular after formulas as the punctuation is very often missing or incorrect (e.g. sentence finishing with a ','). In particular in the Appendix where the punctuation is chaotic.

References: many references are incomplete (lacks conference/journal)

---

> ### Author Response · Authors · 2022-07-31
> **Answers to questions and concerns**
>
> We really appreciate your detailed and professional review. Here are our responses to the questions and comments:
>
> - Q1: GeoLIP comes from the geometric ideas applied in this work. **C3. How is quantitative geometry related** in the general response section provides more discussion.
>
> - Q2: We are not very sure about the question. The result from NGeoLIP is the returned value from the Matlab solver to the corresponding SDP program, independent of the approximation result, i.e., even if we were unaware of the approximation guarantee, the experimental result would still stay the same. The SDP relaxation means maximizing over a larger space, so the maximum is also larger than the unrelaxed program, which provides an upper bound, but we do not usually know how good this upper bound is. The approximation guarantee indicates that this upper bound is not much larger than the maximum of the original program (see appendix A2 in the updated paper). If this does not answer your question, we are happy to provide more information.
>
> - Q3: In this case, the strong duality holds. The natural relaxation programs in section 3 are strictly feasible because the identity matrix is a positive definite solution, so Slater’s condition holds. We add this fact as a remark (Remark 4.2) in the updated paper. We have run a few hundred pairs of SDP programs during the development of this work, and all the dual pairs produce the same results, up to a negligible numerical difference from the solver’s numerical tolerance. (Each MNIST network produces 10 pairs of $\ell_\infty$ programs and 10 pairs of $\ell_2$ programs because there are 10 predictions.)
>
> - Q4: Extending the SDPs in section 4 to larger networks is easy when it comes to composing the SDP, because our program reasoning technique is compositional, and it is straightforward to write the CNN or residual connections as quadratic constraints. We can then apply Shor’s relaxation scheme to derive an SDP. The bottleneck is solving the SDP.
> There are two ways of improving the SDP solving: The first is to exploit the chordal sparsity [1,2], to decompose a large SDP constraint to a few smaller ones. The second is to implement faster matrix operations, such as multiplication and inverse, with respect to the SDP constraint matrix block structure, as proposed in [3].
>
> - Convention: The upper indices are necessary when we introduce multiple layer networks in section 4.2, paragraph *Multi-layer extension*. If we used lower indices, it would be confusing to distinguish which matrix and vector we are referring to.
>
> - ref for the majoration of KG and its link to eq (5): Krivine showed that $K_G\leq \frac{\pi}{2 \ln(1+\sqrt{2})} = 1.782…$, and Braverman et al. showed that $K_G<  \frac{\pi}{2 \ln(1+\sqrt{2})} $. One can view the SDP relaxation in eq (5) as the sum of the inner product of vectors, whose Euclidean norms are 1. Because $X\succeq 0$, $X=M*M^T$, where $M\in \mathbb{R}^{(n+m)\times d}$ for some $d\geq 1$. $u_i$ are the first $n$ row vectors of $M$, and $v_j$ are the last $m$ row vectors of $M$, so $X_{ij} = \langle u_i, v_j\rangle$ for $i\leq n$ and $j\geq n+1$. $X_{kk}=1$ means $||u_k||=1$ for $1\leq k\leq n$ and $||v_k||=1$ for $n+1\leq k\leq n+m$. Thus, $K_G$ in Theorem 3.1 is the approximation guarantee. Notice that if $d=1$ in $M$'s dimension, the SDP coincides with the combinatorial problem, because the inner product degenerates to the multiplication of two scalars. So the SDP relaxation can be viewed as a continuous relaxation of a discrete problem, and the inequality quantifies this geometric transformation.
>
> - In all generality, the dual of ℓ∞ is ill-defined: we agree with this if the underlying space is infinite-dimensional, and we need to impose extra conditions on the functional. Is this the concern?
>
> We have edited the paper as advised, and appreciate these suggestions.
>
> [1]Matthew Newton and Antonis Papachristodoulou. 2021. Exploiting Sparsity for Neural Network Verification. In Proceedings of the 3rd Conference on Learning for Dynamics and Control (Proceedings of Machine Learning Research, Vol. 144, PMLR, 715–727) https://proceedings.mlr.press/v144/newton21a.html
>
> [2]Anton Xue, Lars Lindemann, Alexander Robey, Hamed Hassani, George J. Pappas, and Rajeev Alur. 2022. Chordal Sparsity for Lipschitz Constant Estimation of Deep Neural Networks. https://doi.org/10.48550/arxiv.2204.00846
>
> [3] Patricia Pauli, Niklas Funcke, Dennis Gramlich, Mohamed Amine Msalmi, and Frank Allgöwer. 2022. Neural network training under semidefinite constraints. https://doi.org/10.48550/ARXIV.2201.00632

---

> > ### Comment · Reviewer_TC4r · 2022-08-08
> > **Thank you for your answer**
> >
> > Thank you for clarifying these points. I read and appreciated your answer. I have also read the other questions raised by all reviewers and the following discussions.
> > I do not have any further question for now.

---

### Author Response · Authors · 2022-07-30
**General Response**

We thank all reviewers. Some reviews point out our paper is not well organized. Because our paper has lots of content and we have a relatively small page budget, our presentation is not perfected and some details are omitted. We update the paper (**in the supplementary material**) as the reviews suggest. Due to the page limit, we keep some of the edits in the appendix. Some organizational changes in the main text have not been updated yet. We will update the main paper if it is accepted and we have an extra page. Here we provide an intuitive explanation and the overall contribution of our work, and then address some common concerns.

### Overview
Our goal is roughly to upper bound the operator-norm of all possible gradients. For a given input, we know its activation pattern, so we can compute the operator-norm of the gradient at that point. Note that the activation pattern of an input is not only decided by the weight matrix, but also the bias term. That’s why if any optimization program does not utilize the bias term, it cannot produce the true Lipschitz constant. However, analyzing the activation pattern for all inputs is infeasible, and we know that the derivative at each hidden node has a fixed range, so we can upper bound the true Lipschitz constant with the Formal Global Lipschitz constant (FGL), which is the maximum of the gradient operator norm assuming all activation patterns are independent and possible even though some of them are not in reality.

Note that in this work, we have two relaxations: the first is to relax the true Lipschitz constant to FGL, and then use SDP to relax the FGL estimation.

On two-layer networks, we reduce the FGL estimation to the mixed-norm problem, by devising a cube transformation technique. This allows us to apply well-studied theoretical techniques (natural SDP relaxations) and results, and also hints that on two-layer networks, it is unlikely to provide more precise estimations on FGLs within polynomial time.

On multi-layer networks, we interpret the derivative at a hidden node as $\frac{\Delta \sigma(y)}{\Delta y}$, and restate the FGL optimization problem as a quadratic program. We then apply Shor’s relaxation to relax the quadratic program to an SDP. Notice that our reasoning for transforming a neural network to a quadratic program is compositional, and not restricted to DNN or $\ell_p$-norm perturbations. This is a novel program-analysis technique for data-independent perturbation analysis of neural networks, and has more applications.

 ### Contributions

1. Theoretics: We connect the FGL estimation problem with the mixed-norm problem, which establishes the theoretical aspect of the FGL estimation such as its computational hardness and approximability. Moreover, we reveal the hidden connections between [1] and LipSDP-neuron, and also hint that [1] and LipSDP-neuron are likely optimal within their application scope, so LipSDP-network might be wrong, which was confirmed by [2].

2. Practicality: We give a program-analysis interpretation to LipSDP-neuron, which allows LipSDP-neuron applicable to broader settings. Our data-independent perturbation reasoning is compositional, and can be extended to other analyses or structures. In particular, we only need to write the neural-network property of interest as a quadratic program.

3. Relevance: We are one of the few known techniques that can work for both $\ell_2$ and $\ell_\infty$ perturbations. We hope that our high-level idea can benefit future works on transferring techniques between metric spaces. There are also several optimization works [3,4] studying the structure proposed in LipSDP. However, given [2] refuted LipSDP, these works might appear ungrounded. Our work reestablishes the correctness of LipSDP-neuron, and shows that [3,4] potentially have broader impacts.

[1] Aditi Raghunathan, Jacob Steinhardt, and Percy Liang. 2018. Certified Defenses against Adversarial Examples. In International Conference on Learning Representations. https://openreview.net/forum?id=Bys4ob-Rb

[2]Patricia Pauli, Anne Koch, Julian Berberich, Paul Kohler, and Frank Allgöwer. 2022. Training Robust Neural Networks Using Lipschitz Bounds. IEEE 31 Control Systems Letters 6 (2022), 121–126. https://doi.org/10.1109/LCSYS.2021.3050444

[3]Matthew Newton and Antonis Papachristodoulou. 2021. Exploiting Sparsity for Neural Network Verification. In Proceedings of the 3rd Conference on Learning for Dynamics and Control (Proceedings of Machine Learning Research, Vol. 144, PMLR, 715–727) https://proceedings.mlr.press/v144/newton21a.html

[4]Anton Xue, Lars Lindemann, Alexander Robey, Hamed Hassani, George J. Pappas, and Rajeev Alur. 2022. Chordal Sparsity for Lipschitz Constant Estimation of Deep Neural Networks. https://doi.org/10.48550/arxiv.2204.00846

---

> ### Author Response · Authors · 2022-07-30
> **General Response (Common concerns)**
>
> ### C1.Notational confusion for forward networks
>  We attempted to use a simplified presentation of the network, but it was incorrect and created confusion. The forward network notation was not directly related to our algorithms or theory, and our intention was to specify the dimensions of matrices in the network from the notation. For line 103, there is no diag. For two-layer networks, we intend the network to be $f(x) = u \sigma (Wx + b_1)$, and using $y$ to denote the values of $\sigma’$ at the hidden layer; and similarly for multi-layer network in section 4.2. We apologize for this confusion, and have already updated the paper.
>
> ### C2. Applicability of the SDPs
> We encode the FGL estimation as a compositional quadratic program, and then apply Shor’s relaxation to the program. It is easy to see that we can extend this analysis to other activations such as ELU and sigmoid, just as LipSDP; and structures like convolutional layers and skip connections. We can use the SDP to reason any neural-network property that can be written in the compositional quadratic program form. For example, if we want to know the output sensitivity to a single input change, we can substitute the perturbation constraint in the program with the single-input-perturbation encoding.
>
> ### C3. How is quantitative geometry related
> GeoLIP comes from the fact that we are realizing geometric techniques to estimate the Lipchitz constant, for example, our approximation results come from the geometric inequalities: Grothendieck (for $\infty\rightarrow 1$) and little Grothendieck (for $\infty\rightarrow 2$)  inequalities [5], and their computational implications [6].
>
> The specific example of the quantitative geometric principle considered in this paper is how we transfer LipSDP-neuron to the $\ell_\infty$-perturbation. We interpret LipSDP-neuron as Shor’s relaxation for a quadratic program, and the quadratic program exactly encodes the underlying perturbation geometry, neural network computation, and the Lipschitz objective. To transfer the LipSDP-neuron to the $\ell_\infty$-setting, we only need to encode a different geometry, and the rest remains the same. Our work is an example that deep learning is connected to metric geometry, and we believe that this perspective can bring more theoretical and mathematical tools to the investigation of neural networks, especially when the underlying space is equipped with different metrics.
>
> ### C4. Multi-layer network theoretical guarantee
> We do not know the theoretical guarantees for multi-layer networks. As discussed in Appendix B, if we interpret the FGL estimation from the polynomial-optimization perspective, this can be viewed as the tensor-norm problem, which the theory community does not know whether is easy or hard in the approximability sense. However, because of the low-rank structure, the FGL estimation can be easier than the general tensor-norm problem. Our perturbation analysis in Section 4 can be viewed as exploiting this structure in practice. We leave the theoretical guarantee for multi-layer networks as an open problem.
>
> ### C5. Sampling
> Notice that sampling can only give a lower bound on the Lipschitz constant, while we need an upper bound as in most works [1,2,4]. We use sampling as a sanity check to ensure that the SDP method is at least sound and indeed provides an upper bound of the FGL. For example, LipSDP was shown to fail to produce an upper bound by [2]. Note that it is not the case that if the number is smaller in the table, the measurement is better. In the extreme case, if we only sampled one input and computed its gradient operator norm, we would have a very small number. However, it is useless and does not reflect the stability of the neural network.
>
> ### C6. Intervals or vertices on the hypercube
> In practice, the interval and vertex representations of cubes do not make any difference in the paper. The algorithm and approximation results remain the same. The difference is the MAXSNP result (whether we can build a reduction to a combinatorial problem) and whether we can provide a ground truth of FGL in the evaluation, i.e., the brute-force method for hypercube vertices. To avoid unnecessary technical details, we change the paper slightly: we use [0, 1] instead of {0, 1} for ReLU’s derivative, which is also consistent with the forward perturbation analysis (section 4); and point out that for the two-layer case, [0,1] is equivalent to the {0, 1} in the maximization problems considered in the paper. The equivalence is provided in Appendix A1.
>
> [5]Holden Lee, Assaf Naor, and Kiran Vodrahalli. 2016. Metric embeddings and geometric inequalities (Lecture Notes). https://web.math.princeton.edu/~naor/mat529.pdf
>
> [6]Vijay Bhattiprolu, Euiwoong Lee, and Madhur Tulsiani. 2022. Separating the NP-Hardness of the Grothendieck Problem from the Little-Grothendieck Problem. In 13th Innovations in Theoretical Computer Science Conference (ITCS 2022). https://doi.org/10.4230/LIPIcs.ITCS.2022.22

---

### Author Response · Authors · 2022-08-05
**General Response**

We appreciated the thoughtful review, and are available to answer any questions.

---

### Meta-Review · Area_Chair_sz6R · 2022-08-26

**Recommendation:** Accept
**Confidence:** Certain

**Metareview:**

All the reviewers agree that the paper is novel and interesting and it should be accepted. Please take into account the reviewers' comments while preparing the camera-ready version, particularly the ones on the clarity of the paper.

**Award:**

No

---

### Decision · Program_Chairs · 2022-09-14

Accept